# WHEN REASONING MEETS ITS LAWS

## ABSTRACT

Despite the superior performance of Large Reasoning Models (LRMs), their reasoning behaviors are often counterintuitive, leading to suboptimal reasoning capabilities. To theoretically formalize the desired reasoning behaviors, this paper presents the Laws of Reasoning (LoRe), a unified framework that characterizes intrinsic reasoning patterns in LRMs. We first propose *compute law* with the hypothesis that the reasoning compute should scale linearly with question complexity. Beyond compute, we extend LoRe with a supplementary *accuracy law*. Since the question complexity is difficult to quantify in practice, we examine these hypotheses by two properties of the laws, *monotonicity* and *compositionality*. We therefore introduce LoRe-Bench, a benchmark that systematically measures these two tractable properties for large reasoning models. Evaluation shows that most reasoning models exhibit reasonable monotonicity but lack compositionality. In response, we develop an effective finetuning approach that enforces compute-law compositionality. Extensive empirical studies demonstrate that better compliance with compute laws yields consistently improved reasoning performance on multiple benchmarks, and uncovers synergistic effects across properties and laws.

## 1 INTRODUCTION

Large Reasoning Models (LRMs) such as OpenAI o1 (Jaech et al., 2024) have demonstrated unprecedented progress in approaching human-like reasoning capabilities. Despite their strong performance on solving complex problems, even powerful LRMs exhibit *abnormal behaviors* that deviate from typical human reasoning patterns. Human generally adapt their thinking based on problem complexity (Newell et al., 1972). In contrast, as illustrated in Fig. 1, DeepSeek-R1 (Guo et al., 2025) tends to generate longer reasoning but with a lower accuracy on a simpler sub-problem[1].

We also identify this unexpected phenomenon across a wide range of reasoning models, as shown in Fig. 5. This is primarily because researchers generally overlook the high variability of Chain-of-Thought (CoT) (Wei et al., 2022) data during the training phase. These CoT data are heuristically curated by human annotators or generated through online rollout (Schulman et al., 2017; Shao et al., 2024), rarely constrained by explicit rules, *e.g.*, how much thinking budget to allocate for a given problem (Wu et al., 2025). Hence, the current training paradigm fails to guide models toward an optimal thinking strategy. It will lead to *inefficient allocation of computation*—either overthinking (Chen et al., 2024b; Sui et al., 2025) or underthinking (Su et al., 2025; Yang et al., 2025; Wang et al., 2025), which in turn harms the performance (Stechly et al., 2024; Zhou et al., 2025a).

To overcome this limitation, one line of work focuses on adaptive *post-training* techniques, including supervised fine-tuning with variable-length CoT (Aggarwal & Welleck, 2025; Team et al., 2025). Another line of work modulates reasoning at *test time* (Muennighoff et al., 2025; Fan et al., 2025; Zhang et al., 2025b). While many attempts have been made to control reasoning, existing approaches primarily rely on ad-hoc heuristics and still behave undesirably in our studies. Therefore, beyond empirical methods, several key challenges remain: **(Q1)** *Can we theoretically formalize model reasoning to ensure desirable behavior?* (Section 2) **(Q2)** *How can we evaluate whether popular LRMs follow these proposed principles?* (Section 3) **(Q3)** *Does enforcing these principles further improve general reasoning capabilities?* (Section 4, 5)

To fill this gap, we introduce **the Laws of Reasoning** (LoRe), which systematically formalize the relationship between complexity and model reasoning behaviors in LRMs. The LoRe framework

---

[1]For each problem, we generated multiple samples to account for randomness.

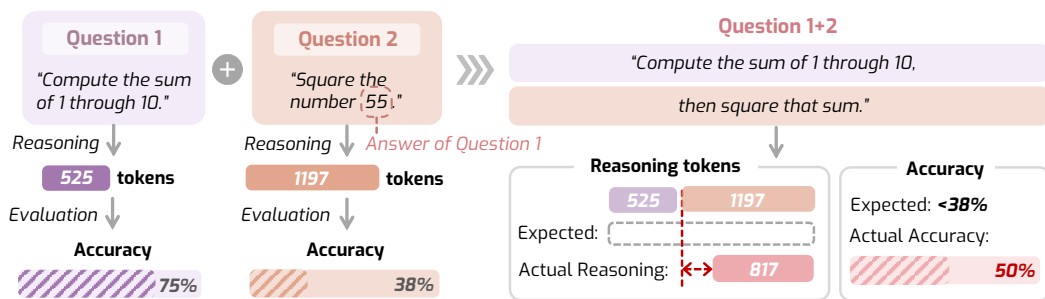

Figure 1: **Illustrative example with DeepSeek-R1** on (a) a summation question, (b) a squaring question, and (c) their composition ("sum, then square"). The model allocates ~300 *more* reasoning tokens to solve the squaring question than to the composite question, with a 12.5% *accuracy drop*. The mismatch with human reasoning reveals an abnormal reasoning pattern present in current LRMs.

comprises a core *compute law* and a complementary *accuracy law*. Given the practical challenges of measuring these hypotheses, the two fundamental laws are approximated via two tractable properties of optimal reasoning models, monotonicity and compositionality.

We then evaluate whether current LRMs follow the laws by developing LORE-BENCH, **a comprehensive benchmark** that examines monotonicity and compositionality in LRMs. While LORE-MONO is a curated benchmark across diverse domains for monotonicity, LORE-COMPO is constructed from MATH500 (Lightman et al., 2023) to measure compositionality. Our evaluation shows that current models exhibit reasonable monotonicity but *lack compositionality*, even for competitive baselines.

In response, we propose **a simple yet effective fine-tuning approach** to enforce the compute-law compositionality. From validation experiments, we present three key insights: (1) the compositionality of reasoning compute can be greatly improved with *simple fine-tuning* approach; (2) Enforcing compositionality *generally leads to better reasoning capability*; (3) *Synergistic effects* emerge, yielding broader improvements across different properties and laws.

## 2 THE LAWS OF REASONING

We introduce the Laws of Reasoning (LORE), a unified framework that formalizes the relationship between question complexity and model reasoning behaviors. Specifically, we focus on two key aspects, reasoning compute and accuracy, which are fundamental to understanding how models scale, generalize, and allocate computation budget when solving complex problems. Section 2.1 formulates the key concepts of reasoning. In Section 2.2, we present the central *compute law*, with a hypothesis that the reasoning budget should scale proportionally with question complexity. In Section 2.3, we introduce the complementary *accuracy law*, which posits that overall accuracy should decay exponentially with increasing complexity. See Fig. 2 for an illustration of the overall framework.

### 2.1 PROBLEM FORMULATION

**Notation.** Let $x \in \mathcal{X} \subseteq \mathcal{V}^*$ denote a question, where $\mathcal{V}^*$ is the space of finite-length sequences over a vocabulary $\mathcal{V}$. Let $M_\theta \in \mathcal{M}$ denote an *autoregressive* large reasoning model. LRMs adopts the *thinking-then-answering* paradigm (Guo et al., 2025; Abdin et al., 2025; Comanici et al., 2025), where the model $M_\theta$ first generates a reasoning chain $r \in \mathcal{R} \subseteq \mathcal{V}^*$ with probability $p_\theta(r \mid x)$ and then an answer $y \in \mathcal{Y} \subseteq \mathcal{V}^*$ with probability $p_\theta(y \mid x, r)$. We assume a fixed decoding strategy by default and denote the model's output by $o = (r, y) \in \mathcal{O} \subseteq \mathcal{V}^*$. We define the composition of two questions $x_1$ and $x_2$ as their concatenation with a connector prompt $c \in \mathcal{V}^{*2}$, *i.e.*, $x_1 \oplus x_2 = \text{concat}(x_1, c, x_2)$.

**Definition 1** (Complexity). Let a *unit-cost primitive step* denote a single valid transition of a fixed deterministic Turing machine (Turing et al., 1936), and let $\tau$ be any finite sequence of primitive steps with length $\ell(\tau) \in \mathbb{N}$. Let $v(x, \tau) \in \{0, 1\}$ be a binary verifier that accepts $(x, \tau)$ if and only if $\tau$ is a valid solution sequence for $x$. The *complexity* of $x \in \mathcal{X}$ is

$$\kappa(x) \triangleq \min\{ \ell(\tau) : v(x, \tau) = 1 \} \in \mathbb{N} \cup \{\infty\},$$

---

[2]One example of $c$ can be "Answer the following questions in order: Q1. {Q1}\nQ2. {Q2}".

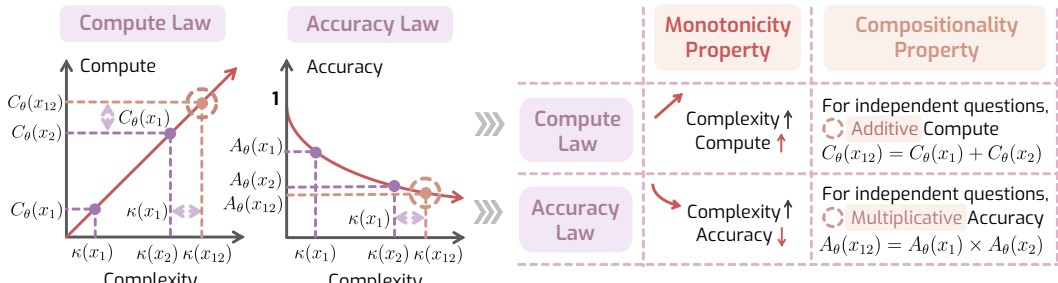

Figure 2: **Overview of the LORE Framework.**

with $\kappa(x) = \infty$ if no valid solution sequence exists.

Here the complexity refers to the *minimal number of unit-cost primitive steps*. Conceptually, $\kappa(x)$ can be well-defined via a binary verifier and a fixed deterministic Turing machine. However, computing $\kappa(x)$ is generally intractable, as verifying the *minimal* solution length requires a global search over a potentially exponential space.

For a given model, its test-time reasoning compute is directly proportional to the number of reasoning tokens generated. We therefore quantify reasoning compute as follows.

**Definition 2** (Reasoning Compute). The *reasoning compute* on question $x$ is defined as the expected number of reasoning tokens generated by the model: $C_\theta(x) \triangleq \mathbb{E}_{r \sim p_\theta(\cdot|x)}[\ell(r)]$, where $\ell(r)$ denotes the length (in tokens) of the reasoning chain $r$.

**Definition 3** (Reasoning Accuracy). The *reasoning accuracy* is defined as the probability that the model, when generating a reasoning chain and an answer given input $x$, produces a final answer that matches the ground truth. Formally, $A_\theta(x) \triangleq \mathbb{E}_{(r,y) \sim p_\theta(\cdot|x)} [\mathbf{1}\{\text{ans}(y) = a^\star(x)\}] \in [0, 1]$. where $a^\star(x)$ denotes the correct answer to $x$, and $\text{ans}(y)$ extracts the final answer from $y$.

## 2.2 COMPUTE LAW

We hypothesize that, if a reasoning model allocates its reasoning compute efficiently, the amount of compute is expected to scale proportionally with complexity in approximation, *i.e.*, $C_\theta(x) \propto \kappa(x)$:

**Hypothesis 1** (Compute Law). For an optimal reasoning model $M_\theta$ and a question $x$ with complexity $\kappa(x)$, there exist $\alpha_\theta > 0$ with,

$$C_\theta(x) = \alpha_\theta\, \kappa(x) + o(\kappa(x)),$$

for some $\alpha_\theta > 0$ that depends only on $M_\theta$ and the decoding strategy. $o(\kappa(x))$ denotes a small systematic overhead that is sublinear, *i.e.*, $o(\kappa)/\kappa \to 0$ when $\kappa \to \infty$.

Specifically, the $o(\kappa(x))$ term captures the introductory and transition tokens during the reasoning process. These tokens generally constitute a very small portion of the overall reasoning and can therefore be ignored in practice.

**Two Tractable Alternative Properties as Proxies.** As discussed in Definition 1, the complexity $\kappa(x)$ is difficult to measure in practice. Consequently, empirically validating the linear relationship is nontrivial, as it would require known complexity values for individual questions. To address this, we adopt two tractable properties as empirical proxies for studying the laws: *monotonicity* and *compositionality*. These properties offer two key advantages: (i) they are tractable to verify without access to the exact value of $\kappa(x)$. Monotonicity relies only on relative comparisons between questions, while compositionality tests whether compute is additive over independent question pairs; (ii) they are theoretically sufficient to imply the proposed compute law (Proposition 1).

**Property 1** (Compute-Complexity Monotonicity). For $x_1, x_2 \in \mathcal{X}$, the reasoning compute is *monotonically non-decreasing* with complexity:

$$\kappa(x_1) \leq \kappa(x_2) \implies C_\theta(x_1) \leq C_\theta(x_2).$$

**Definition 4** (Independence). For $x_1, x_2 \in \mathcal{X}$, $x_1$ and $x_2$ are *independent* if the complexity of their composition is additive, *i.e.*, $\kappa(x_1 \oplus x_2) = \kappa(x_1) + \kappa(x_2)$.

In practice, since the exact complexity values are difficult to obtain, we define independence operationally. Suppose each question $x \in \mathcal{X}$ is associated with a set of mathematical concepts[3] $\mathcal{S}(x) \subseteq \mathcal{S}$ relevant to solving it. We consider two questions $x_1$ and $x_2$ to be independent if their concept sets are disjoint, i.e., $\mathcal{S}(x_1) \cap \mathcal{S}(x_2) = \varnothing$.

**Property 2** (Compute-Complexity Compositionality). For $x_1, x_2 \in \mathcal{X}$, if $x_1$ and $x_2$ are independent, their composite $x_1 \oplus x_2$ exhibits *additive compute*:

$$C_\theta(x_1 \oplus x_2) = C_\theta(x_1) + C_\theta(x_2) + o(\kappa(x_1) + \kappa(x_2)),$$

where the sublinear terms accounts for systematic overhead in the reasoning process (as assumed in Hypothesis 1). Therefore, the reasoning compute is *approximately additive*:

$$C_\theta(x_1 \oplus x_2) \approx C_\theta(x_1) + C_\theta(x_2).$$

**Discussion.** Intuitively, these properties are motivated by two basic principles: (i) more complex questions naturally require more reasoning; (ii) Two independent sub-questions involve no overlapping reasoning, so the total compute is the sum of solving each one individually. In the next proposition, we state informally that these properties imply the compute law (Hypothesis 1); a formal proof is provided in Appendix D. These tractable properties thus offer a practical means to evaluate whether current LRMs follow the compute law.

**Proposition 1.** Under certain conditions, if a reasoning model $M_\theta$ satisfies compute-complexity monotonicity and compositionality, then its reasoning compute $C_\theta(x) \propto \kappa(x)$ for $x \in \mathcal{X}$.

### 2.3 BEYOND COMPUTE: ACCURACY LAW

Following Definition 1, suppose a question requires solving $\kappa(x)$ unit-cost primitive steps. If each step succeeds independently with a fixed probability and all steps must succeed for the final answer to be correct, then the overall accuracy is expected to decrease exponentially with $\kappa(x)$. This intuition motivates the following formulation of the accuracy law:

**Hypothesis 2** (Accuracy Law). For an optimal reasoning model $M_\theta$ and a question $x$ with complexity $\kappa(x)$, when $0 < A_\theta(x) \leq 1$, there exists $\lambda_\theta \geq 0$ with,

$$A_\theta(x) = \exp\big(-\lambda_\theta \kappa(x)\big).$$

Equivalently, $\log A_\theta(x) \propto -\kappa(x)$, where $\lambda_\theta \geq 0$ is the decay rate.

Similar to the compute law, we assume that the reasoning accuracy for the optimal reasoning model $M_\theta$ also satisfies two fundamental properties: *monotonicity* and *compositionality*.

**Property 3** (Accuracy-Complexity Monotonicity). For $x_1, x_2 \in \mathcal{X}$, the reasoning accuracy is *monotonically non-increasing* with complexity:

$$\kappa(x_1) \leq \kappa(x_2) \implies A_\theta(x_1) \geq A_\theta(x_2).$$

**Property 4** (Accuracy-Complexity Compositionality). For $x_1, x_2 \in \mathcal{X}$, if $x_1$ and $x_2$ are independent, their composite $x_1 \oplus x_2$ exhibits *multiplicative accuracy*:

$$A_\theta(x_1 \oplus x_2) = A_\theta(x_1) \cdot A_\theta(x_2).$$

**Discussion.** These properties are motivated by two basic principles: (i) more complex questions tend to have lower accuracy; (ii) for two independent questions with accuracies $p_1$ and $p_2$ (e.g., $p_1 = 0.8$, $p_2 = 0.7$), the probability of correctly answering both should be $p_1 \cdot p_2$ (e.g., 0.56). We state below that these properties imply the accuracy law, with a formal proof provided in Appendix D.

**Proposition 2.** Under certain conditions, if a reasoning model $M_\theta$ satisfies accuracy-complexity monotonicity and compositionality, then its reasoning accuracy $\log A_\theta(x) \propto -\kappa(x)$ for $x \in \mathcal{X}$.

## 3 DO CURRENT LRMs FOLLOW THE LAWS?

In this section, we systematically evaluate whether current LRMs follow our proposed *reasoning laws*. Specifically, we introduce LORE-BENCH, a two-fold benchmark that leverages two tractable properties, monotonicity and compositionality, to examine LRMs.

---

[3]For example, concepts may come from Calculus (*e.g.*, derivatives), Algebra (*e.g.*, group theory), or Discrete mathematics (*e.g.*, logic).

Figure 3: **Question Generation of LORE-MONO.** For each seed question, we generate 30 variants with increasing complexity. Specifically, variant $N$ applies the update rules $N$ times to compute the answer, so the question complexity increases monotonically with $N$.

### 3.1 LORE-MONO

Evaluating the monotonicity property in Property 1 or Property 3 requires comparing the complexity of arbitrary question pairs. However, due to its definition via minimal solution length, complexity is inherently difficult to quantify in practice. As a result, existing benchmarks are not suited for such analysis. To address this challenge, we construct LORE-MONO, a synthetic benchmark where questions are carefully curated and validated to follow known complexity orderings, allowing us to systematically assess the monotonicity of reasoning compute and accuracy.

**(1) Seed Question Curation.** We select four domains that require extensive reasoning—*math*, *science*, *language*, and *code*—and curate 10 diverse seed questions for each. A seed question defines a problem template shared across its variants.

**(2) From Seed Questions to Variants.** As shown in Fig. 3, for each seed question, we create a series of variants (30 in total) that become increasingly complex by requiring more steps to reach the final answer. For example, variant 1 requires one matrix operation, variant 2 requires two, and variant 30 requires thirty, with the identical operation applied repeatedly. By design, a larger number of steps directly corresponds to higher complexity. Note that this construction does not assume or require each operation to use the same compute.

**(3) Program-based Generation and Manual Verification.** All variants are generated through Python scripts to ensure correctness and scalability. To prevent unintended shortcuts such as periodic patterns, we manually verify each seed question and review sampled variants. We provide detailed seed questions and variants for each domain in Appendix E.

We use the Spearman correlation coefficients $\rho \in [-1, 1]$ to measure how the variant index, which directly determines the constructed question's complexity, relates to two quantities: reasoning compute and log accuracy. A high correlation with reasoning compute indicates that compute grows monotonically with complexity (Property 1), while a negative correlation with log accuracy indicates that accuracy tends to degrade as complexity increases (Property 3).

### 3.2 LORE-COMPO

In contrast, assessing compositionality is more straightforward: it only requires taking any two independent questions as sub-questions and constructing their composition. We build LORE-COMPO from MATH500 (Lightman et al., 2023), where each question is labeled by subject (e.g., Algebra, Geometry). Specifically, we randomly sample a pair of questions $(x_1, x_2)$ from *distinct pre-defined subjects* to ensure independence, and concatenate them into a composite question $x_{12}$. Each original question is used at most once, yielding 250 triplets, each with two sub-questions and their composition: $\mathcal{D}_{\text{LoRe-Compo}} = \{(x_1^{(i)}, x_2^{(i)}, x_{12}^{(i)})\}_{i=1}^{250}$. Recall that for a function $f_\theta(\cdot)$ (either $C_\theta(\cdot)$ or $\log A_\theta(\cdot)$), compositionality implies that $f_\theta(x_{12}) \approx f_\theta(x_1) + f_\theta(x_2)$. We therefore quantify the degree to which a model follows this property using the mean absolute deviation (MAD):

$$\text{MAD}_f = \sum_{(x_1, x_2, x_{12}) \in \mathcal{D}_{\text{LoRe-Compo}}} \left| f_\theta(x_{12}) - \left( f_\theta(x_1) + f_\theta(x_2) \right) \right|$$

A smaller MAD indicates stronger adherence to the compositionality property. However, MAD is scale-dependent. To address this, we adopt the Normalized MAD (nMAD):

$$\text{nMAD}_f = \frac{\text{MAD}_f}{S_f}, \quad S_f = \sum_{(x_1, x_2, x_{12}) \in \mathcal{D}_{\text{LoRe-Compo}}} |f_\theta(x_1) + f_\theta(x_2)|.$$

### 3.3 FINDINGS AND ANALYSIS

**Evaluation Setups.** We examine 6 LRMs on LORE-MONO and LORE-COMPO: four standard models— DeepSeek-R1-Distill (Qwen-1.5B, Qwen-7B, Llama-8B) (Guo et al., 2025) and Phi-4-mini-reasoning (Xu et al., 2025a)—and two models that apply reasoning length control, Thinkless-1.5B-RL-DeepScaleR (Fang et al., 2025) and AdaptThink-7B-delta0.05 (Zhang et al., 2025a). For each question, we sample 8 outputs per model with a fixed decoding temperature (0.6 for the DeepSeek family and 0.8 for the Phi-4 family from their technical reports) and a maximum length of 20480 tokens. For LORE-MONO, at each variant index we first average reasoning compute[4] and log accuracy across the 40 questions, and then compute the Spearman correlation.

Table 1: **Monotonicity Results on LORE-MONO.** We examine whether reasoning compute and log accuracy of 6 popular LRMs satisfy the monotonicity property across four domains. Spearman correlations are reported for reasoning compute and log accuracy. *Lang.* stands for *Language*.

| Size | Models | Reasoning Compute ↑ | | | | | Log Accuracy ↓ | | | | |
|---|---|---|---|---|---|---|---|---|---|---|---|
| | | Math | Science | Lang. | Code | All | Math | Science | Lang. | Code | All |
| 1.5B | DeepSeek-R1-1.5B | 0.861 | 0.910 | -0.346 | 0.151 | **0.875** | -0.795 | -0.864 | -0.210 | -0.487 | **-0.868** |
| | Thinkless-1.5B | 0.943 | 0.961 | 0.648 | 0.794 | **0.976** | -0.951 | -0.934 | -0.556 | -0.539 | **-0.960** |
| 3.8B | Phi-4-mini | 0.980 | 0.973 | 0.936 | 0.922 | **0.988** | -0.965 | -0.802 | -0.911 | -0.822 | **-0.954** |
| 7B | DeepSeek-R1-7B | 0.956 | 0.975 | 0.901 | 0.970 | **0.991** | -0.946 | -0.876 | -0.899 | -0.818 | **-0.978** |
| | AdaptThink-7B | 0.984 | 0.995 | 0.950 | 0.984 | **0.995** | -0.963 | -0.949 | -0.904 | -0.888 | **-0.972** |
| 8B | DeepSeek-R1-8B | 0.982 | 0.962 | 0.864 | 0.963 | **0.988** | -0.944 | -0.796 | -0.924 | -0.843 | **-0.947** |
| 14B | Nemontron-14B | 0.964 | 0.976 | 0.917 | 0.970 | **0.993** | -0.778 | -0.751 | -0.793 | -0.911 | **-0.913** |
| | DeepSeek-R1-14B | 0.978 | 0.973 | 0.903 | 0.980 | **0.990** | -0.888 | -0.888 | -0.803 | -0.933 | **-0.981** |
| 32B | Sky-T1-32B | 0.988 | 0.982 | 0.711 | 0.874 | **0.967** | -0.939 | -0.876 | -0.783 | -0.860 | **-0.963** |
| 80B | Qwen3-80B-Next | 0.977 | 0.984 | 0.794 | 0.993 | **0.992** | -0.403 | -0.551 | -0.774 | -0.907 | **-0.973** |

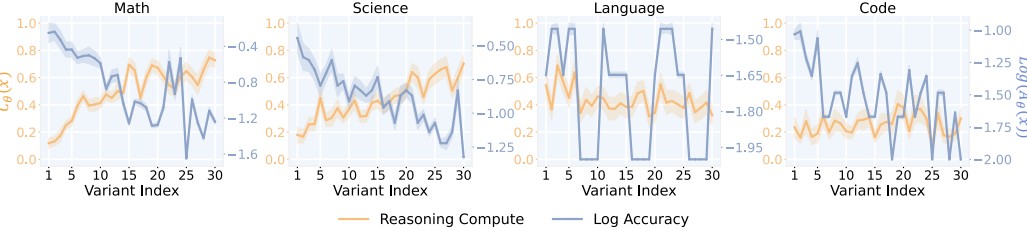

Figure 4: **Visualizations of Monotonicity Results on DeepSeek-R1-1.5B.** For each domain, we plot reasoning compute and log accuracy as a function of variant index. The curves report the mean accuracy across 10 questions series, and the shaded regions denote the standard deviation.

**Current LRMs Largely Satisfy Monotonicity.** On LORE-MONO, all LRMs exhibit a strong positive correlation between reasoning compute and the variant index, which directly reflects question complexity, with most overall Spearman correlations close to 1, as shown in Tab. 1. The only exception is DeepSeek-R1-Distill-Qwen-1.5B, which has the weakest reasoning ability among the six models and yields a lower overall correlation (0.875). As illustrated in Fig. 4, notably, in the language domain its correlation between reasoning compute and complexity is negative (−0.346), while in the

---

[4]We apply max–min normalization to the reasoning compute of each question to prevent any single item from dominating the results.

Table 2: **Compositionality Results on LORE-COMPO.** We calculate nMAD for reasoning compute ($C_\theta$) and log accuracy ($\log A_\theta$).

| Models | nMAD$_{C_\theta}$ ↓ | nMAD$_{\log A_\theta}$ ↓ |
|---|---|---|
| DeepSeek-R1-1.5B | 0.528 | 2.368 |
| Thinkless-1.5B | 0.339 | 0.694 |
| Phi-4-mini | 0.322 | 0.732 |
| DeepSeek-R1-7B | 0.337 | 1.170 |
| AdaptThink-7B | 0.327 | 0.791 |
| DeepSeek-R1-8B | 0.423 | 0.818 |
| DeepSeek-R1-14B | 0.368 | 1.310 |
| Nemotron-14B | 0.393 | 1.073 |
| Qwen3-80B-Next | 0.411 | 1.487 |
| Sky-T1-32B | 0.392 | 1.900 |

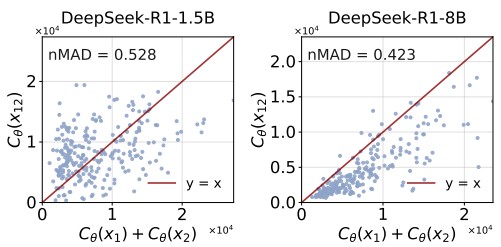

Figure 5: **Visualizations of Compositionality Results on Reasoning Compute.** We plot $C_\theta(x_1 \oplus x_2)$ against $C_\theta(x_1) + C_\theta(x_2)$. Further results are provided in Appendix E.4.

code domain, it is near zero (0.151). This indicates that *in some domains, the reasoning compute for this model does not systematically increase with complexity, and may even decrease*. We provide a case study as additional analysis along with visualization results for other models in Appendix E.2. Meanwhile, most LRMs exhibit a negative correlation between log accuracy and the variant index, as expected. For DeepSeek-R1-Distill-Qwen-1.5B, however, this trend appears noticeably weaker.

**Current LRMs Fail to Exhibit Compositionality.** The nMAD is large for both reasoning compute and log accuracy (Tab. 2), indicating that current LRMs do not satisfy compositionality. Fig. 5 further plots $C_\theta(x_1 \oplus x_2)$ against $C_\theta(x_1) + C_\theta(x_2)$ for two representative LRMs. If an LRM adhered to the compositionality law, most points would align closely with the $y = x$ line. In practice, however, the majority of points deviate substantially. Notably, even models equipped with reasoning length control mechanisms (Thinkless-1.5B and AdaptThink-7B) exhibit considerable deviations, suggesting that such techniques do not inherently promote compositional behavior.

## 4 IMPROVING REASONING VIA ENFORCING COMPOSITIONALITY

In Section 3, we showed that while most LRMs generally satisfy *monotonicity*, they often fail to satisfy *compositionality*. Based on Hypothesis 1, this observation motivates a natural question: *can enforcing compositionality lead to stronger reasoning capacity?* In response, we propose a simple yet effective supervised fine-tuning (SFT) method to promote compositional behavior in LRMs. Importantly, we focus on enforcing compositionality specifically with respect to *reasoning compute*, as it provides a more direct and actionable criterion for selecting supervision examples.[5]

**Proposed Method: `SFT-Compo`** Specifically, let $M_\theta$ be an LRM and $\mathcal{D}_{\text{train}}$ a training dataset. Following the construction in Section 3.2, we select question pairs $(x_1, x_2) \in \mathcal{D}_{\text{train}}$ from distinct categories and form composite questions $x_{12} = x_1 \oplus x_2$. For each triplet $(x_1, x_2, x_{12})$, we sample $K$ model outputs $o = (r, y) \in \mathcal{O}$ from an LRM (either the current model $M_\theta$ or a stronger teacher model) , where $r \in \mathcal{R}$ is a reasoning path and $y \in \mathcal{Y}$ is the corresponding final answer:

$$\{o_1^{(k)} = (r_1^{(k)}, y_1^{(k)})\}_{k=1}^K \text{ for } x_1, \quad \{o_2^{(k)} = (r_2^{(k)}, y_2^{(k)})\}_{k=1}^K \text{ for } x_2, \quad \{o_{12}^{(k)} = (r_{12}^{(k)}, y_{12}^{(k)})\}_{k=1}^K \text{ for } x_{12}.$$

Since compositionality is defined over reasoning paths, among the $K^3$ combinations $(o_1, o_2, o_{12})$, we consider only those where all three reasoning paths $r_1, r_2, r_{12}$ lead to correct answers, and select the combination that best satisfies the compositionality condition:

$$(r_1^*, r_2^*, r_{12}^*) = \underset{r_1, r_2, r_{12}}{\arg\min} |\ell(r_1) + \ell(r_2) - \ell(r_{12})|$$

s.t. $r_1, r_2, r_{12}$ each yielding a correct final answer. (1)

Each triplet thus yields three supervised examples: $(x_1, o_1^*)$, $(x_2, o_2^*)$, and $(x_{12}, o_{12}^*)$, where $o_i^* = (r_i^*, y_i^*)$ with $y_i^*$ the final answer paired with $r_i^*$ in the sampled outputs. Aggregating across all triplets, we construct the compositional supervision dataset $\mathcal{D}_{\text{comp}} =$

---

[5]Accuracy compositionality is not easy to enforce directly, as it does not specify which reasoning path should be selected for supervision.

$\{(x_1, o_1^*),\ (x_2, o_2^*),\ (x_{12}, o_{12}^*) \mid (x_1, x_2) \in \mathcal{D}_{\text{train}}\}$. We then perform SFT on $\mathcal{D}_{\text{comp}}$ to encourage $M_\theta$ to internalize compositional reasoning behavior.

## 5 EXPERIMENTS

We now empirically evaluate `SFT-Compo`, addressing two research questions: (1) whether it effectively enforces compositionality, and (2) whether it further improves the reasoning capacity of LRMs. We also provide additional insightful findings in our analysis.

### 5.1 EXPERIMENTAL SETUP

**Model, Dataset and SFT Recipe.** We evaluate three LRMs: DeepSeek-R1-Distill (Qwen-1.5B, Qwen-7B, Llama-8B) (Guo et al., 2025). We construct a dataset of sub-question and composite-question triplets using a subset of DeepScaler (Luo et al., 2025b). For each question (either sub-question or composite), we use DeepSeek-R1-Distill-Qwen-14B as a stronger teacher model to sample $K = 8$ model outputs. We then construct the compositionality-enforced dataset $\mathcal{D}_{\text{comp}}$ as described in Eqn. 1, which contains approximately 3.9K question-output pairs. We fine-tune each LRM on $\mathcal{D}_{\text{comp}}$ for 5 epochs using a batch size of 16. Additional implementation details are provided in Appendix F.

**Evaluation.** To evaluate compositionality, we use LORE-COMPO. For general reasoning capacity, we consider six benchmarks: GSM8K (Cobbe et al., 2021), MATH500 (Lightman et al., 2023), AIME 2024, AIME 2025 (Mathematical Association of America, 2025), AMC 2023 (AI-MO, 2024), and OlympiadBench (He et al., 2024). We set the maximum generation length to 10240 tokens.

### 5.2 MAIN RESULTS

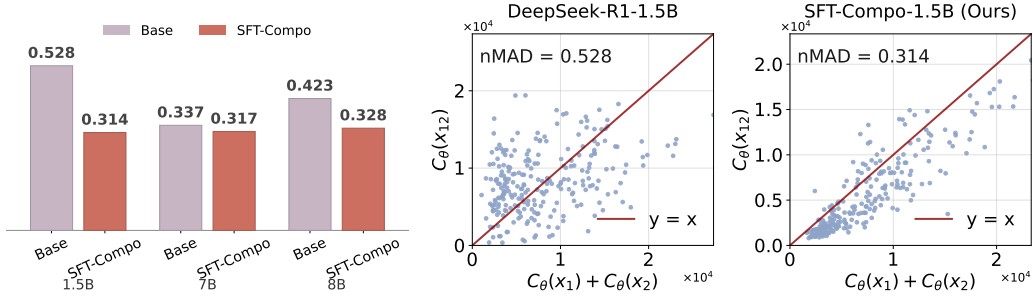

(a) nMAD$_{C_\theta}$ on LORE-COMPO.  (b) Visualizations of Reasoning Compute Compositionality.

Figure 6: **Comparison of Reasoning Compute Compositionality on LORE-COMPO for Base and SFT-Compo models. (a)** `SFT-Compo` consistently achieves a lower nMAD$_{C_\theta}$ across 1.5B, 7B, and 8B models compared to the base model. **(b)** We visualize $C_\theta(x_1 \oplus x_2)$ against $C_\theta(x_1) + C_\theta(x_2)$ for 1.5B models. `SFT-Compo` aligns more closely with the $y=x$ line than the base model.

**Does `SFT-Compo` Effectively Enforce Compositionality Compared to the Base Model?** We compare LRMs before and after SFT using the nMAD of reasoning compute on LORE-COMPO. As shown in Fig. 6a, `SFT-Compo` consistently reduces nMAD compared to the base model. On the 1.5B model, `SFT-Compo` achieves a reduction from $0.528$ to $0.314$ (a 40.5% reduction), and on the 8B model, from $0.423$ to $0.328$ (a 22.5% reduction). We further visualize the results on the 1.5B model in Fig. 6b, where `SFT-Compo` aligns much more closely with the $y=x$ line. Therefore, *the compositionality of reasoning compute can be effectively enforced in a simple manner via* `SFT-Compo`.

**Does Enforcing Compositionality Lead to Stronger Reasoning Capabilities?** As shown in Tab. 3, `SFT-Compo` consistently improves performance across all six benchmarks and all three model sizes. For instance, on the 8B model, it yields a notable gain of $+5.0$ in average Pass@1. To rule out the possibility that performance gains stem solely from leveraging outputs generated by a stronger teacher model, we introduce a control baseline, SFT, which constructs the training dataset by uniformly sampling one correct reasoning path for each question in the triplet:

$$(r_1^*, r_2^*, r_{12}^*) \sim \text{Unif}\left(\{(r_1, r_2, r_{12}) \mid r_1, r_2, r_{12} \text{ each yield a correct final answer}\}\right).$$

Notably, `SFT-Compo` outperforms `SFT` in all cases, showing that the gains are not just from distilling a stronger model but from better compliance with compositionality. This supports our Hypothesis 1—that *stronger models better follow reasoning laws*—and demonstrate that *encouraging compositionality further enhances the reasoning capabilities of LRMs.*

Table 3: **General Reasoning Evaluation Results**. We evaluate `Base` (pre-SFT), `SFT`, and `SFT-Compo` (Ours) on mathematical and science reasoning benchmarks. All numbers report Pass@1 accuracy (%) computed over 8 sampled outputs. $\overline{\text{Pass@1}}$ denotes the average across the six benchmarks. Numbers in **orange** indicate improvements relative to the base model.

| Base Model | Method | MATH | | | | | SCIENCE | $\overline{\text{Pass@1}}$ |
| | | AIME24 | AIME25 | AMC23 | MATH500 | GSM8K | Olympiad | |
|---|---|---|---|---|---|---|---|---|
| **DeepSeek-R1-1.5B** | Base | 18.8 | 20.4 | 59.7 | 71.6 | 81.2 | 33.8 | 47.6 |
| | SFT | $20.4_{+1.6}$ | $21.5_{+1.1}$ | $59.6_{-0.1}$ | $76.4_{+4.8}$ | $81.7_{+0.5}$ | $36.1_{+2.3}$ | $49.3_{+1.7}$ |
| | SFT-Compo (Ours) | $\mathbf{26.2}_{+7.4}$ | $\mathbf{21.7}_{+1.3}$ | $\mathbf{65.0}_{+5.3}$ | $\mathbf{77.6}_{+6.0}$ | $\mathbf{85.1}_{+3.9}$ | $\mathbf{38.7}_{+4.9}$ | $\mathbf{52.4}_{+4.8}$ |
| **DeepSeek-R1-7B** | Base | 36.3 | 27.5 | 79.0 | 86.8 | 91.0 | 48.1 | 61.5 |
| | SFT | $40.0_{+3.7}$ | $32.5_{+5.0}$ | $80.4_{+1.4}$ | $88.0_{+1.2}$ | $\mathbf{91.6}_{+0.6}$ | $48.4_{+0.3}$ | $63.5_{+2.0}$ |
| | SFT-Compo (Ours) | $\mathbf{43.3}_{+7.0}$ | $\mathbf{33.2}_{+5.7}$ | $\mathbf{80.6}_{+1.6}$ | $\mathbf{88.8}_{+2.0}$ | $\mathbf{91.6}_{+0.6}$ | $\mathbf{50.5}_{+2.4}$ | $\mathbf{64.7}_{+3.2}$ |
| **DeepSeek-R1-8B** | Base | 28.3 | 22.9 | 71.9 | 76.4 | 86.5 | 40.9 | 54.5 |
| | SFT | $30.4_{+2.1}$ | $24.2_{+1.3}$ | $75.2_{+3.3}$ | $82.6_{+6.2}$ | $88.0_{+1.5}$ | $44.7_{+3.8}$ | $57.5_{+3.0}$ |
| | SFT-Compo (Ours) | $\mathbf{31.3}_{+3.0}$ | $\mathbf{29.2}_{+6.3}$ | $\mathbf{76.9}_{+5.0}$ | $\mathbf{83.0}_{+6.6}$ | $\mathbf{89.5}_{+3.0}$ | $\mathbf{46.8}_{+5.9}$ | $\mathbf{59.5}_{+5.0}$ |

### 5.3 SYNERGISTIC EFFECT ANALYSIS

**Enforcing Compositionality in Reasoning Compute Improves Its Monotonicity.** Recall from Section 3.3 that DeepSeek-R1-Distill-Qwen-1.5B initially exhibits relatively weak monotonicity in reasoning compute. Fig. 7a shows `SFT-Compo` significantly improves this property, increasing the overall Spearman correlation from $0.875$ to $0.977$. Specifically, in the code domain, it rises from $0.151$ to $0.914$. This indicates that enforcing compositionality can implicitly enhance monotonicity.

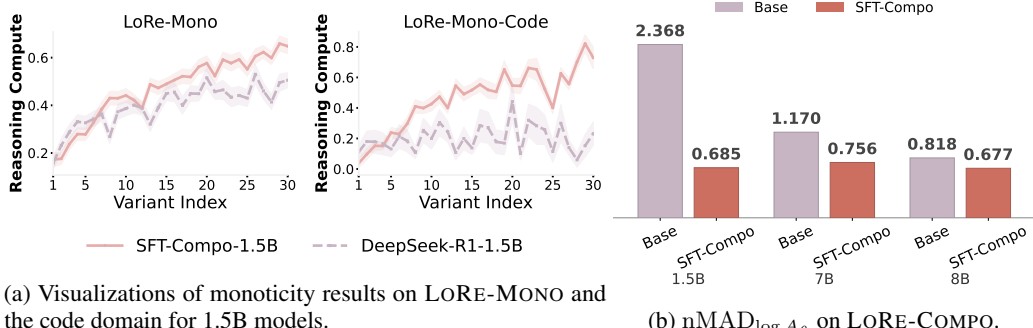

(a) Visualizations of monoticity results on LORE-MONO and the code domain for 1.5B models.

(b) $\text{nMAD}_{\log A_\theta}$ on LORE-COMPO.

Figure 7: **Synergistic Effects Among Different Reasoning Properties and Laws. (a)** Enforcing compositionality in reasoning compute improves its monotonicity. **(b)** Enforcing compositionality in reasoning compute also improves the compositionality of log accuracy, measured by $\text{nMAD}_{\log A_\theta}$.

**Enforcing Compositionality in Reasoning Compute Improves Compositionality in Accuracy.** Interestingly, though `SFT-Compo` is designed to enhance compositionality in reasoning compute, it improves the compositionality of log accuracy. Fig. 7b shows that the nMAD of log accuracy drops from $2.368$ to $0.685$ on the 1.5B model (a $71.1\%$ reduction), and from $1.170$ to $0.756$ on the 7B model (a $35.4\%$ reduction). This suggests a possible interplay among different reasoning laws.

## 6 RELATED WORK

LRMs have emerged as a family of foundation models (Wiggins & Tejani, 2022). Since the advent of OpenAI o1 (Jaech et al., 2024), the "thinking-then-answering" paradigm has been widely adopted, with notable follow-ups such as DeepSeek-R1 and Phi-4-Reasoning (Abdin et al., 2024; Guo et al.,

2025; Qwen Team, 2025). Our framework builds upon the contemporary paradigm of adaptive reasoning, wherein the model's reasoning budget is dynamically controlled either through post-training interventions (Luo et al., 2025a; Zhou et al., 2025b) or at test time (Muennighoff et al., 2025; Xu et al., 2025b; Zhang et al., 2025b). Specifically, one line of work explores post-training techniques that modulate when and how long a model should reason (Chen et al., 2024a; Yong et al., 2025), while another frontier focuses on dynamically adjusting reasoning behavior during inference (Qiao et al., 2025; Liu & Wang, 2025). Refer to Appendix B for additional related work.

## 7 CONCLUSIONS

As a comprehensive study from theoretical hypotheses to empirical validation, we advance a theoretical perspective grounded in human reasoning for improving reasoning in LRMs. We hope LoRE can inspire more potential strategies that guide models toward their optimal paradigms of thinking.

## ETHICS STATEMENT

This work does not raise any known ethical concerns.

## REPRODUCIBILITY STATEMENT

Our code repository is available at `https://anonymous.4open.science/r/Reasoning_laws-02F5/`. In Section 3.3 and Section 5.1, we provide a detailed description of the experimental setup, including dataset, models, training and evaluation procedures. Additional implementation details can be found in Appendix F.1.

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

## A  LLM Usage

LLMs were used solely for language polishing.

## B  Additional Related Work

**Large Reasoning Models.**  Large Reasoning Models (LRMs) have emerged as a family of foundation models (Wiggins & Tejani, 2022). Since the advent of OpenAI o1 (Jaech et al., 2024), this "thinking-then-answering" paradigm has been widely adopted. Notably, o1-like Reasoning Models can solve increasingly complex reasoning problems through elaborate reasoning chains (Wei et al., 2022; Yao et al., 2023; Besta et al., 2024). Numerous efforts replicating o1's success include DeepSeek-R1 and Phi-4-Reasoning (Abdin et al., 2024; Guo et al., 2025; Qwen Team, 2025). Despite impressive progress, the internal mechanisms and behavioral patterns of reasoning in LRMs remain underexplored. Shojaee et al. (2025) take a step in this direction by examining reasoning through the lens of problem complexity, though their analysis is limited to a constrained puzzle-solving setting.

**Reasoning Length Control.**  Our framework builds upon the contemporary paradigm of adaptive reasoning, in which the reasoning budget of the model is controlled either during post-training (Luo et al., 2025a; Zhou et al., 2025b) or at test time (Muennighoff et al., 2025; Xu et al., 2025b; Zhang et al., 2025b). One line of work develops post-training techniques that modulate when and how long a model should reason (Chen et al., 2024a; Yong et al., 2025). This is achieved through two primary strategies: one involves supervised fine-tuning on variable-length CoT with concise yet sufficient reasoning (Aggarwal & Welleck, 2025; Team et al., 2025); the other utilizes RL through length penalty (Zhang et al., 2025a; Fang et al., 2025; Liu et al., 2025). Beyond these, another frontier involves implementing dynamic control over reasoning during inference. For example, some approaches allocate inference budget via confidence (Qiao et al., 2025; Liu & Wang, 2025), while others employ a secondary controller to modulate (Li et al., 2025).

## C  Limitations and Future Work

We acknowledge several limitations. First, our LORE-MONO currently includes only 40 seed questions in total. Expanding its topic diversity and coverage is an important direction for future work. Second, we operationalize independence through disjoint sets of mathematical concepts. Although this proxy is not rigorous, it is motivated by the practical difficulty of formalizing independence between questions in an actionable and general way. We leave more refined treatments of independence to future work. Finally, due to budget constraints, we focus on strong open-source LRMs, as evaluating closed-source models would require substantial additional cost.

## D  Proofs and Corollaries

We first restate Proposition 1 and Proposition 2 formally and provide a complete proof, along with corresponding corollaries.

**Proposition 1** (Formal Version)**.** Fix a question space $\mathcal{X}$, a *complexity* map $\kappa : \mathcal{X} \to \mathbb{N} \cup \{\infty\}$, and a *reasoning* compute map $C_\theta : \mathcal{X} \to \mathbb{R}_{\geq 0}$. Let $\oplus$ be a binary composition operator. For $m \geq 3$ and jointly independent $x_1, \ldots, x_m$, define $x_1 \oplus \cdots \oplus x_m$ by a fixed bracketing (e.g. right-associated). Define

$$\mathcal{X}_{\text{fin}} := \{x \in \mathcal{X} : \kappa(x) < \infty\}, \qquad K := \kappa(\mathcal{X}_{\text{fin}}) \subseteq \mathbb{N}.$$

All assumptions below are imposed on $\mathcal{X}_{\text{fin}}$.

**(A1) Monotonicity.** If $\kappa(x) \leq \kappa(y)$ then $C_\theta(x) \leq C_\theta(y)$.

**(A2) Additivity under composition of independent questions.** If $x, y$ are independent, then

$$\kappa(x \oplus y) = \kappa(x) + \kappa(y), \qquad C_\theta(x \oplus y) = C_\theta(x) + C_\theta(y).$$

**(A3)** For every $u \in K$ and every $m \in \mathbb{N}$, there exist $x_1, \ldots, x_m \in \mathcal{X}_{\text{fin}}$ such that

$$\kappa(x_i) = u \text{ for all } i, \qquad \{x_1, \ldots, x_m\} \text{ is jointly independent.}$$

Consequently, $x_1 \oplus \cdots \oplus x_m$ is valid and

$$\kappa(x_1 \oplus \cdots \oplus x_m) = mu \in K, \qquad C_\theta(x_1 \oplus \cdots \oplus x_m) = \sum_{i=1}^{m} C_\theta(x_i).$$

Then there exists a constant $\alpha_\theta \geq 0$ such that

$$C_\theta(x) = \alpha_\theta\, \kappa(x) \qquad \text{for all } x \in \mathcal{X}_{\text{fin}}.$$

*Proof.* Define an equivalence relation $x \sim y \iff \kappa(x) = \kappa(y)$. By (A1), $\kappa(x) \leq \kappa(y)$ and $\kappa(y) \leq \kappa(x)$ imply $C_\theta(x) \leq C_\theta(y)$ and $C_\theta(y) \leq C_\theta(x)$, hence $C_\theta(x) = C_\theta(y)$ whenever $x \sim y$. Thus there is a well-defined $f : K \to \mathbb{R}_{\geq 0}$ with $f(n) = C_\theta(x)$ for any $x$ such that $\kappa(x) = n$.

Fix $u \in K$ and $m \in \mathbb{N}$. By (A3) choose jointly independent $x_1, \ldots, x_m$ with $\kappa(x_i) = u$. By (A2) and the fixed bracketing,

$$\kappa(x_1 \oplus \cdots \oplus x_m) = mu, \qquad C_\theta(x_1 \oplus \cdots \oplus x_m) = \sum_{i=1}^{m} C_\theta(x_i) = m\, f(u),$$

so

$$f(mu) = m\, f(u) \qquad (\forall\, u \in K,\ \forall\, m \in \mathbb{N}).$$

If $K = \{0\}$ then $C_\theta \equiv 0$ and the claim holds with $\alpha_\theta = 0$. Otherwise take $u, v \in K$ with $u, v > 0$ and let $\ell = \text{lcm}(u, v)$. Then

$$f(\ell) = f\!\left(u \cdot \frac{\ell}{u}\right) = \frac{\ell}{u}\, f(u) \quad \text{and} \quad f(\ell) = f\!\left(v \cdot \frac{\ell}{v}\right) = \frac{\ell}{v}\, f(v),$$

hence $f(u)/u = f(v)/v$, independent of $u, v$. Write this common ratio as $\alpha_\theta \geq 0$. Therefore $f(n) = \alpha_\theta n$ for all $n \in K$, and $C_\theta(x) = f(\kappa(x)) = \alpha_\theta\, \kappa(x)$ for all $x \in \mathcal{X}_{\text{fin}}$. $\qquad\square$

**Corollary D.1** (Asymptotic version with sublinear overhead). *If the compositional compute holds up to a sublinear overhead, i.e., for independent $x, y$,*

$$C_\theta(x \oplus y) = C_\theta(x) + C_\theta(y) + o(\kappa(x) + \kappa(y)),$$

*and the same (A3) assumption holds, then the above proof yields*

$$C_\theta(x) = \alpha_\theta\, \kappa(x) + o\big(\kappa(x)\big) \qquad (\kappa(x) \to \infty).$$

**Proposition 2** (Formal Version). Let $\mathcal{X}_{\text{fin}} = \{x \in \mathcal{X} : \kappa(x) < \infty\}$. Assume the setting and independence notion of Property 3, Property 4, and Assumption (A3). Then there exists $\lambda_\theta \geq 0$ such that for all $x \in \mathcal{X}_{\text{fin}}$ with $0 < A_\theta(x) \leq 1$,

$$A_\theta(x) = \exp\big(-\lambda_\theta\, \kappa(x)\big).$$

*Proof.* Define an equivalence relation $x \sim y$ iff $\kappa(x) = \kappa(y)$. By (A1), if $x \sim y$ then both $\kappa(x) \leq \kappa(y)$ and $\kappa(y) \leq \kappa(x)$ hold, hence $A_\theta(x) \geq A_\theta(y)$ and $A_\theta(y) \geq A_\theta(x)$, so $A_\theta(x) = A_\theta(y)$. Therefore there exists a well-defined map

$$f : K \to (0, 1], \qquad f(n) := A_\theta(x) \text{ for any } x \in \mathcal{X}_{\text{fin}} \text{ with } \kappa(x) = n.$$

Let $g : K \to \mathbb{R}_{\geq 0}$ be $g(n) := -\log f(n)$.

Fix $u \in K$ and $m \in \mathbb{N}$. By (A3), choose jointly independent $x_1, \ldots, x_m$ with $\kappa(x_i) = u$. By (A2) and the fixed bracketing,

$$\kappa(x_1 \oplus \cdots \oplus x_m) = mu, \qquad A_\theta(x_1 \oplus \cdots \oplus x_m) = \prod_{i=1}^{m} A_\theta(x_i) = \big(f(u)\big)^m.$$

Hence

$$g(mu) = -\log A_\theta(x_1 \oplus \cdots \oplus x_m) = m\, g(u) \qquad (\forall u \in K,\ \forall m \in \mathbb{N}). \tag{2}$$

If $K = \{0\}$ then $A_\theta \equiv 1$ and the claim holds with $\lambda_\theta = 0$. Otherwise, let $u, v \in K$ with $u, v > 0$ and set $\ell = \text{lcm}(u, v)$. Applying Eqn. 2 twice gives

$$g(\ell) = g\left(u \cdot \frac{\ell}{u}\right) = \frac{\ell}{u} g(u) \quad \text{and} \quad g(\ell) = g\left(v \cdot \frac{\ell}{v}\right) = \frac{\ell}{v} g(v),$$

so $g(u)/u = g(v)/v$. This ratio is independent of $u, v > 0$ in $K$; denote it by $\lambda_\theta \geq 0$.

For any $n \in K$, if $n = 0$ then $g(n) = 0 = \lambda_\theta n$; if $n > 0$ pick any $u \in K \setminus \{0\}$ and write $n = \frac{n}{u} u$ to get from Eqn. 2 that $g(n) = \frac{n}{u} g(u) = \lambda_\theta n$. Therefore $g(n) = \lambda_\theta n$ for all $n \in K$, i.e. $f(n) = \exp(-\lambda_\theta n)$, and for any $x \in \mathcal{X}_{\text{fin}}$,

$$A_\theta(x) = f\big(\kappa(x)\big) = \exp\big(-\lambda_\theta \kappa(x)\big).$$

$\square$

**Corollary D.2** (Asymptotic version with sublinear coupling). *If for independent $x, y$ the multiplicativity holds up to a sublinear deviation in the exponent,*

$$\log A_\theta(x \oplus y) = \log A_\theta(x) + \log A_\theta(y) + o\big(\kappa(x) + \kappa(y)\big),$$

*and* (A3) *holds, then*

$$\log A_\theta(x) = -\lambda_\theta \kappa(x) + o\big(\kappa(x)\big) \quad (\kappa(x) \to \infty),$$

*equivalently $A_\theta(x) = \exp\big(-\lambda_\theta \kappa(x) + o(\kappa(x))\big)$.*

# E  ADDITIONAL DETAILS AND RESULTS OF LORE-BENCH

## E.1  ADDITIONAL DETAILS OF LORE-MONO

### E.1.1  EXAMPLE SEED QUESTIONS OF LORE-MONO

Here we provide one representative seed question example for each domain.

---

**Math - Example seed question**

Given an integer $n = \{N\}$, consider the order-2 recurrence over integers modulo M with an alternating update rule and a mild nonlinear term. You are given the initial values

$$x_0 = x0, \quad x_1 = x1.$$

We update the sequence one step at a time. Let $t = 1, 2, 3, \ldots, n$ denote the update index, where $t = 1$ is the update that produces $x_2$ from $(x_1, x_0)$. At each update $t$, compute $x_{k+1}$ from $(x_k, x_{k-1})$ using the parity of $t$:
- Define the nonlinear map $\varphi(z) = (z + 1)^2$. (You may reduce intermediate values modulo M at any time.)
- **Odd step (t odd):**

$$x_{k+1} \equiv A x_k + B x_{k-1} + C \varphi(x_k) \pmod{M}.$$

- **Even step (t even):**

$$x_{k+1} \equiv A x_k - B x_{k-1} + C \varphi(x_{k-1}) \pmod{M}.$$

For clarity, the first two updates are:

$$t = 1 : x_2 \equiv A x_1 + B x_0 + C \varphi(x_1) \pmod{M},$$
$$t = 2 : x_3 \equiv A x_2 - B x_1 + C \varphi(x_1) \pmod{M}.$$

Apply exactly $n - 1$ updates starting from $x_0, x_1$ to reach $x_n (n = n)$, and \*\*return $x_n$\*\* as a single non-negative integer in $[0, M - 1]$.
Conventions: - All modular reductions are taken modulo M and return a non-negative remainder.
- The alternating rule depends on the \*\*update index\*\* $t$. - Output only the integer value of $x_n$ (no extra text).

---

864
865

## Code - Example seed question

866
867
868
869
870

You are given runnable Python 3.10 code. Execute it exactly as-is in a clean environment (no extra imports). This is a Code Execution task: run the program, do not rewrite it. The loop counter i is 0-based. Return only the value of ANSWER (no other text, no formatting).
Code:

871

```
N = {N}; s = {init_state!r}

def f(s, i):
    if len(s) == 0:
        return s
    L = len(s); r = (i % L) + 1; s1 = s[r:] + s[:r]
    trans = str.maketrans({'a':'e','e':'i','i':'o','o':'u','u':'a'})
    return s1.translate(trans)

for i in range(N):
    s = f(s, i)

ANSWER = s
```

872
873
874
875
876
877
878
879
880
881
882
883
884
885
886
887
888
889
890
891
892
893
894
895
896

## Science - Example seed question

897
898
899
900

You are modeling a **batch bioreactor** where an enzyme E converts substrate A to product B, but each catalytic event requires a recyclable **cofactor token** C (e.g., NAD*/NADH). Let $A_t, B_t, C_t$ be the nonnegative integer counts of A, B, and C **after** completing tick $t$. You are given fixed initial counts and a regeneration period: - $A_0 = args.A0$, $B_0 = args.B0$, $C_0 = args.C0$ - Regeneration period $k = k$

901
902

For **each discrete tick** $t = 1, 2, \ldots, n$ (with $n = \{N\}$), apply the following **biochemical rule order**:

903
904
905

1) **Reaction (consumes cofactor)** — if both substrate and cofactor are available: - If $A_{t-1} > 0$ **and** $C_{t-1} > 0$, then one catalytic turnover occurs:

906
907

$$A_t = A_{t-1} - 1, \quad B_t = B_{t-1} + 1, \quad C_t = C_{t-1} - 1.$$

- Otherwise, no reaction this tick:

908
909
910

$$A_t = A_{t-1}, \ B_t = B_{t-1}, \ C_t = C_{t-1}.$$

911
912
913

2) **Cofactor regeneration (post-reaction)** — models an external respiratory/oxidative cycle returning the cofactor to its usable form at fixed intervals: - If $t \bmod k = 0$, then **after** the reaction stage:

914

$$C_t \leftarrow C_t + 1.$$

915
916
917

All updates are integer and at most $\pm 1$ per tick ("min/+=1" granularity). **Output** the product count $B_n$ after completing exactly $n = \{N\}$ ticks (i.e., after applying the regeneration rule at tick $n$).

---

**Language - Example seed question**

You are given a letter maze and a number of moves n={N}. The maze is a rectangular grid of letters G with h={H} rows and w={W} columns:

{grid_block}

Start at the cell (r0, c0) = ({R0}, {C0}). Build a string S as you move:
1) First, write down the starting letter G[r0][c0] into S. (This is done before any moves.)
2) Then repeat the following exactly n={step} times (t = 1..n):
• Let (r, c) be your current cell BEFORE moving, and let ch = G[r][c].
• Move one step based on ch (case-insensitive):
− If ch ∈ {a, e, i, o, u} (a vowel): move RIGHT → c ← (c + 1) mod {W}
− Otherwise (a consonant): move DOWN → r ← (r + 1) mod {H}
• After moving to the destination cell (r, c), append its letter G[r][c] to S.
• Now mutate the grid based on ch (the letter you moved FROM):
− If ch is a vowel: cyclically rotate COLUMN c upward by 1.
(Formally, for all i: G[i][c] ← old G[(i + 1) mod H][c].)
− Otherwise (ch is a consonant): cyclically rotate ROW r left by 1.
(Formally, for all j: G[r][j] ← old G[r][(j + 1) mod W].)

Important: The mutation happens AFTER appending G[r][c] to S, and it affects the grid used for the NEXT iteration. Indices are 0-based and the maze wraps around like a torus. Thus, after n moves, S has length n + 1 (because the starting letter was included).

Let k = {K}. Your task is to return the word W made by the LAST k letters of S (in order). Output W as a plain string.

---

### E.1.2 POTENTIAL SHORTCUT FAILURES

A basic requirement of LORE-MONO is that the complexity of question variants increases monotonically with the variant index. However, certain seed questions may violate this requirement. For instance, if answers follow a periodic pattern (*e.g.*, when all even-indexed variants have the answer 1 and all odd-indexed variants have the answer 0), a model could exploit prior patterns to guess the correct answer without performing the intended computation. To ensure benchmark reliability, we manually reviewed all variants and excluded those exhibiting periodic answer patterns.

### E.2 ADDITIONAL RESULTS OF LORE-MONO

### E.2.1 ADDITIONAL VISUALIZATION RESULTS

Visualizations of monotonicity results for the remaining 5 LRMs are provided in Fig. 8.

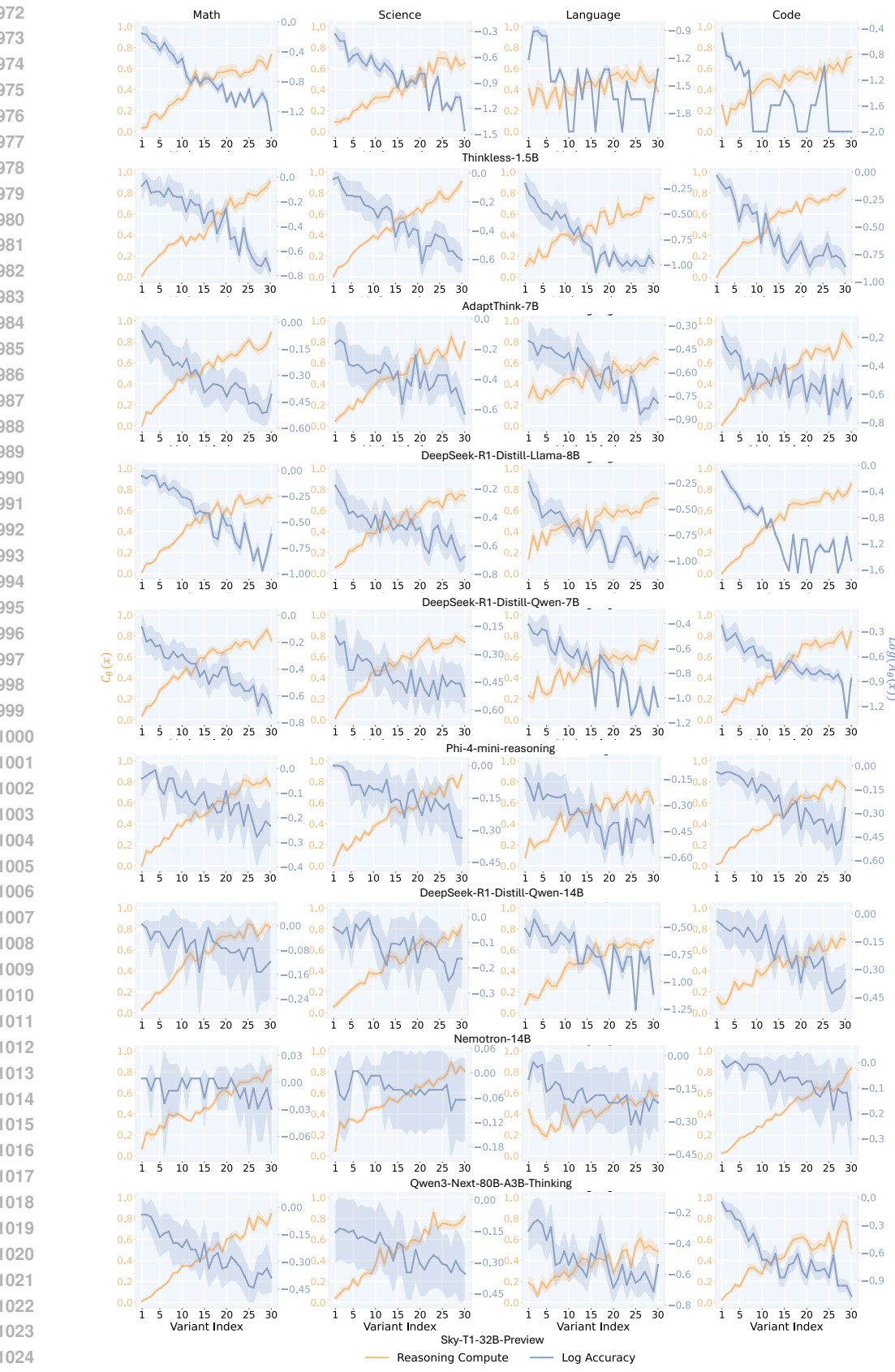

Figure 8: **Visualizations of Monotonicity Results on Remaining LRMs.**

### E.2.2 "Fail to Think" Case Study

We observe that the weakest model, DeepSeek-R1-Distill-Qwen-1.5B, occasionally produces no reasoning tokens on certain code questions. For example, on variant 1 of a seed code question, the generated output is simply <think>\n</think>, corresponding to a reasoning length of 0.

---

**"Fail to think" - Example model output**

<think>\n</think>\n\n To solve this problem, we need to simulate the execution of a given Python code and determine the final value assigned to the variable 'ANSWER'. The code involves string manipulation and state transitions, which we will simulate step by step.\n\n### Approach\n1. **Initialization**:\n - We start with 'N = 1' iteration, 's = \"01938\"', and a function 'f' that processes the string 's'.\n\n2. **Function Definition**:\n - The function 'f' takes a string 's' and processes each character based on its position in the string. The position determines whether the count of '0', '1', '2', or '3' is incremented modulo 5.\n\n3. **Processing Each Character**:\n - For each character in 's', we count the occurrences of '0', '1', '2', and '3'..... (*omitted* ...)

---

### E.3 Additional Details of LoRe-Compo

For all composite questions, a fixed connector links the sub-questions, ensuring they are answered in sequential order. An illustrative example is provided below.

---

**Example composite question**

Answer the following questions in order:
Q1. Jim and Martha are standing together at the corner of a rectangular field. Jim walks diagonally across the field. Martha gets to the same location by walking along its length and width. The field is 300 feet wide and 400 feet long. How many feet less than Martha does Jim walk?
Q2. Find all values of $x$ that satisfy the equation $x = \sqrt{11 - 2x} + 4$.

---

### E.4 Additional Results of LoRe-Compo

Visualizations of compositionality results for the remaining 4 LRMs are provided in Fig. 9.

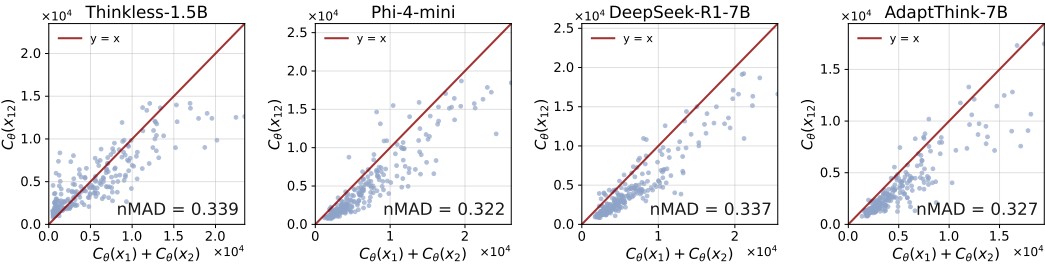

Figure 9: **Visualizations of Compositionality Results on Remaining 4 LRMs.**

## F Experimental Details and Additional Results

### F.1 Implementation Details

Since DeepScaler does not come with predefined categories, we first annotate each question using GPT-4.1-mini to assign it to one of the following categories: Algebra (Prealgebra), Counting &

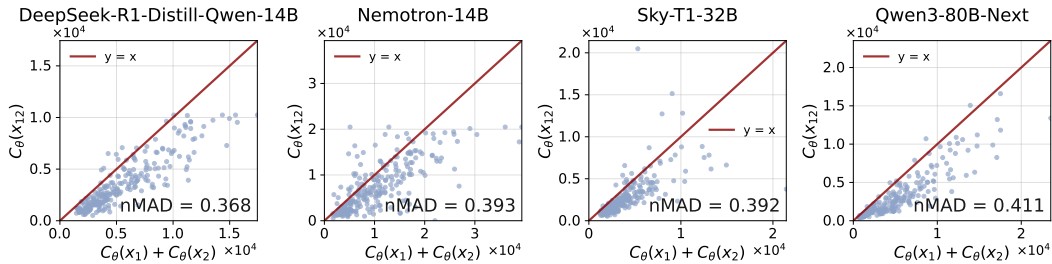

Figure 10: **Visualizations of Reasoning Compute Compositionality on More LRMs.**

Probability, Geometry, Number Theory, or Calculus (Precalculus). Based on these annotations, we construct sub-question and composite-question triplets by pairing questions from different categories.

For SFT, we perform a grid search over learning rates in {1e-6, 5e-6, 5e-5}, using a batch size of 8, gradient accumulation of 2, and a warmup ratio of 0.

### F.2 ADDITIONAL EXPERIMENTAL RESULTS

In Fig. 11, we further compare the reasoning compute compositionality of DeepSeek-R1-Distill-Qwen-7B and DeepSeek-R1-Distill-Llama-8B before and after SFT-Compo. With SFT-Compo, the nMAD decreases, and the results align more closely with the $y = x$ line compared to their base counterparts.

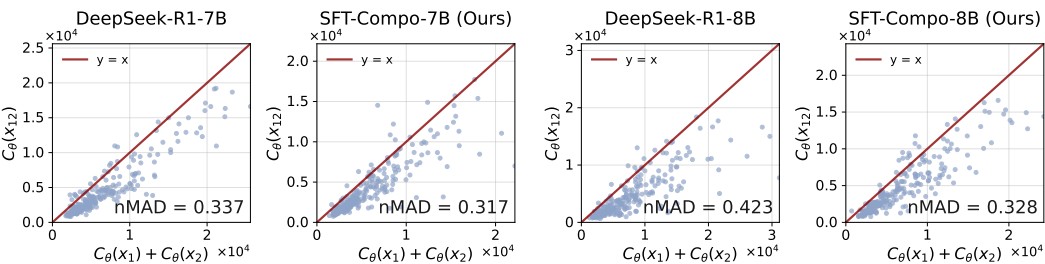

Figure 11: **Visualizations of Reasoning Compute Compositionality on DeepSeek-R1-Distill-Qwen-7B and DeepSeek-R1-Distill-Llama-8B.**

# G    QUALITATIVE EXAMPLES OF LORE-COMPO

We present an example generated of DeepSeek-Distill-1.5B and after SFT-Compo. For DeepSeek-Distill-1.5B, we report metrics for the question pair and the composite question with accuracies $A_\theta(x_1) = 1.0$, $A_\theta(x_2) = 1.0$, $A_\theta(x_{12}) = 0.25$. Reasoning compute is $C_\theta(x_1) = 2177.25$, $C_\theta(x_2) = 3546.50$, and $C_\theta(x_{12}) = 732.13$, showing that the model severely underthinks on the composite question and compromises the accuracy. After SFT-Compo, the composite accuracy improves to $A_{\tilde\theta}(x_{12}) = 0.875$ with reasoning compute as $C_{\tilde\theta}(x_{12}) = 4300.00$. We provide detailed reasoning processes of the two models below.

---

**Example composite question**

Q: Answer the following questions in order:
Q1. Pat is to select six cookies from a tray containing only chocolate chip, oatmeal, and peanut butter cookies. There are at least six of each of these three kinds of cookies on the tray. How many different assortments of six cookies can be selected? (Note that cookies of the same type are not distinguishable.)
Q2. Let $f(x) = x^{10} + 5x^9 - 8x^8 + 7x^7 - x^6 - 12x^5 + 4x^4 - 8x^3 + 12x^2 - 5x - 5$.
Without using long division (which would be horribly nasty!), find the remainder when $f(x)$ is divided by $x^2 - 1$.

---

**Example reasoning process (DeepSeek-R1-1.5B)**

Q1. To determine the number of different assortments of six cookies, I need to consider the number of ways to select six cookies from three types: chocolate chip, oatmeal, and peanut butter. Since the cookies of the same type are indistinguishable, the order of selection doesn't matter.
I can model this problem using combinations with repetition. The formula for the number of ways to choose $k$ items from $n$ types is given by:

$$\binom{n + k - 1}{k}$$

In this case, $n = 3$ (types of cookies) and $k = 6$ (cookies to select). Plugging in the values:

$$\binom{3 + 6 - 1}{6} = \binom{8}{6}$$

Calculating $\binom{8}{6}$ gives:

$$\binom{8}{6} = \frac{8!}{6!(8 - 6)!} = 28 \quad \text{[Correct Answer]}$$

Therefore, there are 28 different assortments of six cookies that can be selected.
Q2. To find the remainder when $f(x)$ is divided by $x^2 - 1$ without using long division, I can utilize the Remainder Theorem. The Remainder Theorem states that the remainder of a polynomial $f(x)$ divided by $x - a$ is $f(a)$. However, since the divisor here is $x^2 - 1$, which factors into $(x - 1)(x + 1)$, I need to evaluate $f(x)$ at the roots of $x^2 - 1$, which are $x = 1$ and $x = -1$.
**[To be continued]**

> **Example reasoning process (DeepSeek-R1-1.5B)**
>
> The remainder will be a linear polynomial of the form $R(x) = ax + b$. To find $a$ and $b$, I can set up two equations by plugging in $x = 1$ and $x = -1$ into $f(x)$: 1. $f(1) = a(1) + b$ 2. $f(-1) = a(-1) + b$
> Calculating $f(1)$:
>
> $$f(1) = 1^{10} + 5(1)^9 - 8(1)^8 + 7(1)^7 - 1(1)^6 - 12(1)^5 + 4(1)^4 - 8(1)^3 + 12(1)^2 - 5(1) - 5$$
>
> $$f(1) = 1 + 5 - 8 + 7 - 1 - 12 + 4 - 8 + 12 - 5 - 5$$
>
> $$f(1) = -1$$
>
> Calculating $f(-1)$:
>
> $$f(-1) = (-1)^{10} + 5(-1)^9 - 8(-1)^8 + 7(-1)^7 - 1(-1)^6 - 12(-1)^5 + 4(-1)^4$$
> $$-8(-1)^3 + 12(-1)^2 - 5(-1) - 5$$
>
> $$f(-1) = 1 - 5 - 8 - 7 - 1 + 12 + 4 + 8 + 12 + 5 - 5$$
>
> $$f(-1) = 13$$
>
> Now, setting up the equations: 1. $a + b = -1$ 2. $-a + b = 13$
> Solving these equations: Adding both equations: $2b = 12 \Rightarrow b = 6$ Substituting $b = 6$ into the first equation: $a + 6 = -1 \Rightarrow a = -7$
> Thus, the remainder $R(x)$ is:
>
> $$R(x) = -7x + 6 \quad \text{[Incorrect Answer]}$$
>
> </think>

1242
1243
1244
1245
1246
1247
1248
1249
1250
1251
1252
1253
1254
1255
1256
1257
1258
1259
1260
1261
1262
1263
1264
1265
1266
1267
1268
1269
1270
1271
1272
1273
1274
1275
1276
1277
1278
1279
1280
1281
1282
1283
1284
1285
1286
1287
1288
1289
1290
1291
1292
1293
1294
1295

### Example reasoning process (DeepSeek-R1-1.5B SFT-Compo)

Okay, so I have these two questions to solve. Let me start with the first one.

**Q1. Pat is to select six cookies from a tray containing only chocolate chip, oatmeal, and peanut butter cookies. There are at least six of each of these three kinds of cookies on the tray. How many different assortments of six cookies can be selected? (Note that cookies of the same type are not distinguishable.)**

Hmm, okay. So Pat is selecting six cookies, and the tray has three types: chocolate chip, oatmeal, and peanut butter. Each type has at least six cookies, so there's no shortage. Since the cookies of the same type are indistinct, we're dealing with combinations with repetition, I think.

I remember that the formula for combinations with repetition is C(n + k - 1, k), where n is the number of types and k is the number of items to choose. In this case, n is 3 (since there are three types of cookies) and k is 6 (since Pat is selecting six cookies). So plugging into the formula, it should be C(3 + 6 - 1, 6) = C(8, 6).

Wait, let me double-check. So, combinations with repetition: if order doesn't matter and items are indistinct except for type, yes, that's the case here. So, the formula is indeed C(n + k - 1, k). So, n=3, k=6, so 3 + 6 -1 = 8, so C(8,6). Alternatively, C(8,2) since combinations are symmetric, which is 28. So is that the answer?

But wait, let me think again. Since each type has at least six cookies, and we're selecting six, does that affect anything? Because if each type has at least six, then when we select six, the maximum number we can take from any one type is six, but since we're only selecting six, the maximum from any type is six, but since we're selecting six in total, the maximum from any type is six, but the minimum is zero.

But in this case, since we have at least six of each, so when we choose six, the number of ways is the same as the number of non-negative integer solutions to the equation:

x1 + x2 + x3 = 6, where x1, x2, x3 are the number of chocolate chip, oatmeal, and peanut butter cookies respectively.

Which is exactly the stars and bars problem, so C(6 + 3 -1, 3 -1) = C(8,2) = 28.

So yeah, the answer is 28. That seems right.

**Q2. Let $f(x) = x^{10} + 5x^9 - 8x^8 + 7x^7 - x^6 - 12x^5 + 4x^4 - 8x^3 + 12x^2 - 5x - 5.$**

Without using long division (which would be horribly nasty!), find the remainder when $f(x)$ is divided by $x^2 - 1$. Okay, so polynomial division. But $f(x)$ is a degree 10 polynomial, and we're dividing by $x^2 - 1$, which is also degree 2. So the remainder should be of degree less than 2, so it should be a linear polynomial, right? So it can be written as ax + b, where a and b are constants.

**[To be continued]**

To find a and b, we can use the Remainder Theorem. Since $x^2 - 1$ factors as $(x - 1)(x + 1)$, the remainder when f(x) is divided by $x^2 - 1$ is the same as the remainder when f(1) and f(-1) are computed, and then we can set up equations to solve for a and b.

Wait, let me recall: if we have f(x) divided by $(x - c)(x - d)$, the remainder is R(x) = e x + f. So, to find R(x), we can evaluate f(c) and f(d), which gives us two equations:

$f(c) = ec + f \quad f(d) = ed + f$

Then, solving these two equations gives us e and f. In our case, the divisor is $x^2 - 1$, which is $(x - 1)(x + 1)$. So, c =1 and d = -1.

Therefore, we can compute f(1) and f(-1), set up the equations:

$f(1) = a(1) + b = a + b$

$f(-1) = a(-1) + b = -a + b$

Then, solving these two equations will give us a and b.

So, let me compute f(1) and f(-1).

First, f(1):

$f(1) = (1)^{10} + 5 * (1)^9 - 8 * (1)^8 + 7 * (1)^7 - (1)^6 - 12 * (1)^5 + 4 * (1)^4 - 8 * (1)^3 + 12 * (1)^2 - 5 * (1) - 5$

Simplify each term:

$1 + 5 - 8 + 7 - 1 - 12 + 4 - 8 + 12 - 5 - 5$

Let me compute step by step: Start with 1.

$1 + 5 = 6 \quad 6 - 8 = -2 \quad -2 + 7 = 5 \quad 5 - 1 = 4 \quad 4 - 12 = -8 \quad -8 + 4 = -4$
$-4 - 8 = -12 \quad -12 + 12 = 0 \quad 0 - 5 = -5 \quad -5 - 5 = -10$

So $f(1) = -10$. Similarly, compute $f(-1)$:

$f(-1) = (-1)^{10} + 5 * (-1)^9 - 8 * (-1)^8 + 7 * (-1)^7 - (-1)^6 - 12 * (-1)^5 + 4 * (-1)^4 - 8 * (-1)^3 + 12 * (-1)^2 - 5 * (-1) - 5$

Simplify each term:

$(-1)^{10} = 1 \quad 5 * (-1)^9 = 5 * (-1) = -5 \quad -8 * (-1)^8 = -8 * (1) = -8$
$7 * (-1)^7 = 7 * (-1) = -7 \quad -(-1)^6 = -1 \quad -12 * (-1)^5 = -12 * (-1) = 12$
$4 * (-1)^4 = 4 * (1) = 4 \quad -8 * (-1)^3 = -8 * (-1) = 8 \quad 12 * (-1)^2 = 12 * (1) = 12 \quad -5 * (-1) = 5$

-5 remains as is.

So let's write all terms: $1 - 5 - 8 - 7 - 1 + 12 + 4 + 8 + 12 + 5 - 5$

Let me compute step by step:

Start with 1.

$1 - 5 = -4 \quad -4 - 8 = -12 \quad -12 - 7 = -19 = \quad -19 - 1 = -20$
$-20 + 12 = -8 \quad -8 + 4 = -4 \quad -4 + 8 = 4 \quad 4 + 12 = 16 \quad 16 + 5 = 21 \quad 21 - 5 = 16$

So $f(-1) = 16$. Therefore, we have: $f(1) = a + b = -10 \quad f(-1) = -a + b = 16$

So now, we have the system of equations: 1) $a + b = -10$ 2) $-a + b = 16$. Let me solve this.

Add equations 1 and 2: $(a + b) + (-a + b) = (-10) + 16$

Simplify: $0a + 2b = 6$ So $2b = 6 => b = 3$. Then, plug b=3 into equation 1: $a + 3 = -10 => a = -13$. So the remainder is $ax + b = -13x + 3$

Therefore, the remainder when f(x) is divided by $x^2 - 1$ is $-13x + 3$. Let me double-check my computations for f(1) and f(-1), just to make sure I didn't make any arithmetic mistakes.

**Recomputing f(1):**

$1 + 5 = 6 \quad 6 - 8 = -2 \quad -2 + 7 = 5 \quad 5 - 1 = 4 \quad 4 - 12 = -8 \quad -8 + 4 = -4$
$-4 - 8 = -12 \quad -12 + 12 = 0 \quad 0 - 5 = -5 \quad -5 - 5 = -10$

Yes, that's correct. **Recomputing $f(-1)$:** $1 - 5 = -4 \quad -4 - 8 = -12 \quad -12 - 7 = -19 \quad -19 - 1 = -20 \quad -20 + 12 = -8 \quad -8 + 4 = -4 \quad -4 + 8 = 4 \quad 4 + 12 = 16 \quad 16 + 5 = 21 \quad 21 - 5 = 16$ Yes, that's correct.

So the equations are: $a + b = -10 \quad -a + b = 16$

Adding them gives $2b = 6 => b = 3$, then a= -13. So the remainder is -13x +3. I think that's correct.

**Final Answer**

Q1: $\boxed{28}$ [Correct Answer]

Q2: $\boxed{-13x + 3}$ [Correct Answer]

</think>