# OpenReview forum: "When Reasoning Meets Its Laws"
_ICLR.cc/2026/Conference — Submitted to ICLR 2026_

### Official Review · Reviewer_pUQ6 · 2025-10-28

**Soundness:** 3
**Presentation:** 3
**Contribution:** 2
**Rating:** 6
**Confidence:** 3

**Summary:**

The paper introduces LORE, a unified theoretical and empirical framework for analyzing and improving reasoning behaviors in Large Reasoning Models (LRMs). The authors formalize two key laws, Compute Law and Accuracy Law. In addition, they propose two tractable empirical proxies, monotonicity and compositionality, and design LORE-BENCH, a benchmark to evaluate whether LRMs follow these properties. The authors then propose SFT-Compo, a simple fine-tuning method to enforce compositionality.

**Strengths:**

1. The idea of defining formal “laws” governing reasoning is original and provides a principled foundation for analyzing LRM behavior.
2. The use of tractable proxies (monotonicity and compositionality) effectively bridges theoretical hypotheses and empirical evaluation.
3. The paper is well presented.

**Weaknesses:**

1. LORE-MONO uses only 40 seed questions, which may not generalize across broader reasoning domains or multimodal contexts.
2. Results might not generalize to stronger closed-source reasoning models.
3. While quantitative metrics are thorough, qualitative case studies or failure analyses would make the behavioral claims more intuitive.
4. The authors did not prove why enforcing compositionality leads to stronger reasoning capabilities.
5. The operational definition of question independence (via disjoint concept sets) is heuristic and may not reflect true cognitive independence.
6. More LRMs could be evaluated.

**Questions:**

1. How robust is the compositionality enforcement to task domain shifts (e.g., from math to commonsense reasoning)?
2. Could the LORE framework be extended to multimodal reasoning tasks (vision-language or embodied reasoning)?

---

> ### Author Response · Authors · 2025-11-27
> **Part 1**
>
> *Thank you for your constructive and supportive reviews that greatly improve our manuscript! Following your review, we conducted **extensive additional experiments**, including evaluations on four large-scale models, additional fine-tuning experiments on Phi4-mini-reasoning, and common sense reasoning. We respond below to your questions and concerns, which we have incorporated into the revised version.*
>
> ***
>
> > **W1**: LoRe-Mono uses only 40 seed questions, which may not generalize across broader reasoning domains or multimodal contexts.
>
> - **LoRe-Mono covers four major types of reasoning in LRMs and is sufficiently representative for evaluating the monotonicity property in practice.** It comprises meticulously constructed, high-quality questions that span diverse reasoning domains: mathematics, science, language, and code. Each seed question is expanded into 30 variants, yielding **1200** evaluations per model and thus providing a substantial scale for empirical validation. We agree with the reviewer that it can be further improved by expanding the topic diversity and coverage of LoRe-Mono, and we have incorporated this point in the limitation section of our initial submission.
> - **The construction principles of LoRe-Mono can be easily applied to multimodal contexts.** For example, in a visual maze navigation task, a seed question could ask the model to start from a given position, take $N$ moves where each move’s direction and step size are randomly generated within a valid range, and then report the final position. By increasing the number of required moves $N$, we can automatically construct a series of variants with monotonically increasing complexity in the multimodal settings, fully analogous to our language-only design.

---

> > ### Author Response · Authors · 2025-11-27
> > **Part 2**
> >
> > > **W2**: Results might not generalize to stronger closed-source reasoning models.
> >
> > We thank the reviewer for the insightful suggestions. Our evaluation focuses on open-source LRMs due to practical constraints: evaluating our reasoning laws **requires access to the full reasoning process**, which is difficult to obtain for most closed-source commercial systems under current APIs. However, it is technically plausible for the commercial service providers to evaluate their own models based on our framework, and the insights from our paper can be utilized to build better closed-source models.
> >
> > **To further demonstrate the generalization of our conclusions, we conducted additional evaluations on four stronger larger-scale reasoning models**, including OpenReasoning-Nemotron-14B, DeepSeek-R1-Distill-Qwen-14B, Sky-T1-32B-Preview, and Qwen3-Next-80B across LoRe-Mono and LoRe-Compo. Our experiments reveal two important findings,
> >
> > - **All these large-scale models failed to exhibit compositionality.** The $\mathrm{nMAD}$ metric used to evaluate compositionality is large for both reasoning compute and log accuracy across all four models. As shown in Appendix E, most points deviate substantially from the ideal $y=x$ line. This demonstrates that even significantly larger models do not naturally follow the compositionality.
> > - **Stronger models do not necessarily better satisfy monotonicity or compositionality than small-scale models.** These large-scale models often perform worse on compositionality than the smaller models reported in our paper. For instance, Qwen3-Next-80B, despite being significantly larger, performs worst among these models on compositionality for reasoning compute, with an $\mathrm{nMAD}$ of 0.411 compared to 0.337 for DeepSeek-R1-Distill-Qwen-7B (**a 22% degradation in compositionality**). Qwen3-Next-80B also exhibits the poorest monotonicity across all evaluated models on Math and Science domains, with reasoning accuracy failing to decrease systematically with complexity.
> >
> > Detailed experiments and visualizations are provided in **Section 3.3 and Appendix E** of the revised paper, which are marked in blue.
> >
> > **Monotonicity Results on LoRe-Mono** (Spearman correlations $\rho$: $\rho$ closer to 1 for reasoning compute and −1 for accuracy indicate better monotonicity)
> > | Models             | Compute↑      |                              |                 |               |      | Log Accuracy↓  |                               |                 |                |      |
> > |-----------------|------|---------|-------|------|-----|------|---------|-------|------|-----|
> > | |Math|Science| Lang.    | Code     | All  | Math       | Science   | Lang.      | Code       | All  |
> > | DeepSeek-R1-7B (reference)   |0.956 |0.975 |0.901 |0.970 |0.991 |-0.946 |-0.876 |-0.899 |-0.818 |-0.978 |
> > | Nemontron-14B   |0.964 |0.976 |0.917 |0.970 |0.993 |-0.778 |-0.751 |-0.793 |-0.911 |-0.913 |
> > | DeepSeek-R1-14B |0.978 |0.973 |0.903 |0.980 |0.990 |-0.888 |-0.888 |-0.803 |-0.933 |-0.981 |
> > | Qwen3-80B-Next  |0.977 |0.984 |0.794 |0.993 |0.992 |**-0.403** |**-0.551** |-0.774 |-0.907 |-0.973 |
> > | Sky-T1-32B      |0.988 |0.982 |0.711 |0.874 |0.967 |-0.939 |-0.876 |-0.783 |-0.860 |-0.963 |
> >
> > **Compositionality Results on LoRe-Compo** ($\mathrm{nMAD}$ closer to 0 indicate better compositionality)
> > | Models | nMAD (Compute) ↓ | nMAD (Log Accuracy) ↓|
> > |-----------------|------|---------|
> > | DeepSeek-R1-7B (reference) | **0.337** | 1.170 |
> > | Nemontron-14B | 0.393 | 1.073 |
> > | DeepSeek-R1-14B | 0.368 | 1.131|
> > | Qwen3-80B-Next | **0.411** | 1.487 |
> > | Sky-T1-32B | 0.392 | 1.900 |
> >
> > ***
> >
> > > **W3**: Qualitative case studies or failure analyses would make the behavioral claims more intuitive.
> >
> > We thank the reviewer for the insightful suggestions. In the revised version, we add qualitative examples and analysis in **Appendix G**. We present a representative LoRe-Compo example for DeepSeek-R1-Distill-Qwen-1.5B before and after SFT-Compo. For the original model, we report metrics for the question pair and the composite question with accuracies $A_\theta(x_1)=1.0$, $A_\theta(x_2)=1.0$, $A_\theta(x_{12})=0.25$. Reasoning compute is $C_\theta(x_1)=2177$, $C_\theta(x_2)=3547$, and $C_\theta(x_{12} )=732$, indicating that the model **severely underthinks** on the composite question, which compromises accuracy. After applying SFT-Compo, the composite accuracy improves to $A_{\tilde{\theta}}(x_{12})=0.875$ with reasoning compute as $C_{\tilde{\theta}}(x_{12})=4300$. It illustrates how enforcing compositionality leads to more efficient compute allocation and better performance.
> >
> > Further details and the full case are provided in Appendix G, which is marked in blue.

---

> > > ### Author Response · Authors · 2025-11-27
> > > **Part 3**
> > >
> > > > **W4**: The authors did not prove why enforcing compositionality leads to stronger reasoning capabilities.
> > >
> > > We thank the reviewer for raising this point. We agree that it is typically non-trivial to provide a theoretical guarantee. Therefore, our claim is not a formal theorem, but a theoretically motivated *hypothesis* supported by both **intuition** and **empirical evaluation**.
> > >
> > > - **Intuitively, compositionality is an essential property of “ideal” or “well-structured” reasoning.** When the compositionality holds, the model allocates reasoning compute that is approximately additive across independent sub-questions. This leads to more efficient computation allocation: complex questions receive sufficient budget, while simple questions are not given unnecessary extra compute. In contrast, the lack of compositionality is closely related to underthinking and overthinking, both of which have been shown to hurt performance in prior works [1][2][3][4]. From this perspective, enforcing compositionality provides a meaningful structured constraint on the model’s reasoning behavior, which is naturally expected to improve overall reasoning capabilities.
> > > - **We also provide systematic empirical evidence that enforcing compositionality leads to stronger reasoning.** As shown in Table 3, SFT-Compo consistently improves performance across all benchmarks and model sizes. Notably, SFT-Compo outperforms standard SFT in all cases, showing that the gains are not just from distilling a stronger model but from **better compliance with compositionality**.
> > >
> > > [1] Su, Jinyan, et al. "Between underthinking and overthinking: An empirical study of reasoning length and correctness in llms." arXiv preprint arXiv:2505.00127 (2025). \
> > > [2] Wu, Yuyang, et al. "When more is less: Understanding chain-of-thought length in llms." arXiv preprint arXiv:2502.07266 (2025). \
> > > [3] Yang, Wenkai, et al. "Towards thinking-optimal scaling of test-time compute for llm reasoning." arXiv preprint arXiv:2502.18080 (2025). \
> > > [4] Shojaee, Parshin, et al. "The illusion of thinking: Understanding the strengths and limitations of reasoning models via the lens of problem complexity." arXiv preprint arXiv:2506.06941 (2025).
> > >
> > > ***
> > >
> > > > **W5**: The operational definition of question independence (via disjoint concept sets) is heuristic and may not reflect true cognitive independence.
> > >
> > > We thank the reviewer for raising this important point. We agree that true cognitive independence is theoretically difficult to formalize and verify. Therefore, our use of disjoint concept sets as an operational definition serves as **a reasonable and actionable approximation to this theoretically intractable notion**. This limitation is already discussed in the limitation section of our initial submission, and we identified the refinement of operational definitions for independence as an important direction for future work.
> > >
> > > ***
> > >
> > > > **W6**: More LRMs could be evaluated.
> > >
> > > **To further demonstrate the generalization capability of SFT-Compo, we conduct experiments on different model families**, including Phi4-mini-reasoning from Microsoft. Our method still achieves consistent gains over all benchmarks and model families.
> > >
> > > | Base Model | Method | AIME24 | AIME25 | AMC23 | MATH500 | GSM8K | Olympiad | Pass@1 |
> > > |-------------------|------------|--------|--------|-------|--------|-------|----------|--------|
> > > | Phi-4-mini | Base      | 32.5   | 24.6   | 71.9  | 86.4   | 91.4  | 50       | **59.5**   |
> > > |  | SFT       | 40.0   | 26.2   | 73.4  | 87.0   | 91.6  | 49.5     | **61.3**   |
> > > |  | SFT-Compo | 43.7   | 31.7   | 76.9  | 87.0   | 92.8  | 51.2     | **63.9**   |

---

> > > > ### Author Response · Authors · 2025-11-27
> > > > **Part 4**
> > > >
> > > > > **Q1**: How robust is the compositionality enforcement to task domain shifts (e.g., from math to commonsense reasoning)?
> > > >
> > > > To examine whether our compositionality enforcement generalizes beyond the math domain, we evaluated SFT-Compo models trained on DeepScaler directly on a **widely used commonsense reasoning benchmark**, CommonsenseQA.
> > > >
> > > > Specifically, we sampled 500 CommonsenseQA problems and constructed a LoRe-Compo-Commonsense set using the same procedure as in LoRE-Compo. We then measured compositionality using the same metric ($\mathrm{nMAD}_{C _\theta}$, **the lower the better**). The results are shown below:
> > > >
> > > >
> > > > | Base Model         | Before SFT-Compo | After SFT-Compo |
> > > > |--------------------|------------------|-----------------|
> > > > | DeepSeek-R1-1.5B   | 0.475            | **0.407**       |
> > > > | DeepSeek-R1-7B     | 0.481            | **0.450**       |
> > > > | DeepSeek-R1-8B     | 0.437            | **0.381**       |
> > > >
> > > > Across all model sizes, SFT-Compo **consistently reduces** $\mathrm{nMAD}_{C _\theta}$ on commonsense reasoning, indicating **improved compositionality even under domain shift**. These results support that our compositionality enforcement is not limited to math tasks, but transfers to semantically distinct reasoning domains.
> > > >
> > > > ***
> > > >
> > > > > **Q2**: Could the LoRe framework be extended to multimodal reasoning tasks (vision-language or embodied reasoning)?
> > > >
> > > > Yes. The LoRe framework is a general and universal framework that can be naturally applied to multimodal and embodied reasoning tasks.
> > > >
> > > > - **Our formulation only requires a notion of question complexity $\kappa(x)$, reasoning compute $C_\theta(x)$ and accuracy $A_\theta(x)$**, none of which is restricted to language-only inputs. The two key properties, monotonicity and compositionality, remain valid and are theoretically sufficient to imply our reasoning laws in multimodal and embodied settings as well.
> > > > - **For vision-language tasks,** consider a visual reasoning or household environment: more complex tasks, such as solving multi-step visual maze problems, should intuitively require more reasoning or planning steps, which corresponds to the same **monotonicity** we study in the language setting.
> > > > - Similarly, **for the embodied reasoning tasks,** consider an embodied agent that must first navigate to a target location and then manipulate an object. When these two sub-questions are independent, the reasoning compute for the combined “navigate + manipulate” task should be roughly the sum of the reasoning required for solving each individual one, which reflects **compositionality**. The same intuition applies to the accuracy law.

---

### Official Review · Reviewer_HnMv · 2025-10-29

**Soundness:** 2
**Presentation:** 3
**Contribution:** 3
**Rating:** 6
**Confidence:** 3

**Summary:**

This paper introduces the LoRe together with the LoRe-Bench benchmark, demonstrating that large reasoning models generally exhibit monotonicity but fall short in compositionality. To address this limitation, the authors present SFT-Compo, a fine-tuning approach designed to enforce compositional reasoning and thereby enhance overall reasoning performance.

**Strengths:**

- This paper proposes LoRe, which characterizes reasoning laws through monotonicity and compositionality, enriching the evaluation dimensions.

- This paper introduces the SFT-Compo method, which is simple yet effective, significantly improving models’ reasoning compositionality and overall performance.

**Weaknesses:**

1. The experiments in the paper are mainly conducted on closed-source small-scale models, and the current results may be insufficient to demonstrate their applicability to larger-scale or different types of models.

2. In SFT-Compo, the training triplets appear to be generated solely using DeepSeek-R1-Distill-Qwen-14B. Such reliance on a single teacher model may limit data diversity and weaken the robustness of the conclusions. Have the authors considered incorporating multiple teacher models or alternative data sources to enhance generalization?

3. In the General Reasoning Evaluation Results section, the paper only compares the base models, SFT, and SFT-Compo. While this shows that SFT-Compo improves over the original models, it lacks comparisons with other baseline methods (e.g., length-control approaches such as Thinkless or AdaptThink).

**Questions:**

see weaknesses

---

> ### Author Response · Authors · 2025-11-27
> **Part 1**
>
> *Thank you for your constructive and supportive reviews that greatly improve our manuscript! Following your review, we conducted **extensive additional experiments**, including evaluations on four large-scale models, additional finetuning experiments on Phi4-mini-reasoning, multiple teacher models and alternative data sources, and baseline comparisons for Thinkless and AdaptThink. We respond below to your questions and concerns, which we have incorporated into the revised version.*
>
> ***
>
> > **W1**: The experiments in the paper are mainly conducted on closed-source small-scale models, and the current results may be insufficient to demonstrate their applicability to larger-scale or different types of models.
>
> We thank the reviewer for the insightful suggestions. To address this concern, we provide two complementary experiments.
>
> **(A) Additional Evaluation on Larger-Scale Reasoning Models**
>
> We conducted additional evaluations on four more large-scale models, including OpenReasoning-Nemotron-14B, DeepSeek-R1-Distill-Qwen-14B, Sky-T1-32B-Preview, and Qwen3-Next-80B across LoRe-Mono and LoRe-Compo. Our experiments reveal two important findings,
>
> - **All these large-scale models failed to exhibit compositionality.** The $\mathrm{nMAD}$ metric used to evaluate compositionality is large for both reasoning compute and log accuracy across all four models. As shown in Appendix E, most points deviate substantially from the ideal $y=x$ line. This demonstrates that even significantly larger models do not naturally follow the compositionality.
> - **Larger models do not necessarily better satisfy monotonicity or compositionality than small-scale models.** These large-scale models often perform worse on compositionality than the smaller models reported in our paper. For instance, Qwen3-Next-80B, despite being significantly larger, performs worst among these models on compositionality for reasoning compute, with an $\mathrm{nMAD}$ of 0.411 compared to 0.337 for DeepSeek-R1-Distill-Qwen-7B (**a 22% degradation in compositionality**). Qwen3-Next-80B also exhibits the poorest monotonicity across all evaluated models on Math and Science domains, with reasoning accuracy failing to decrease systematically with complexity.
>
> Detailed experiments and visualizations are provided in **Section 3.3 and Appendix E** of the revised paper, which are marked in blue.
>
> **Monotonicity Results on LoRe-Mono** (Spearman correlations $\rho$: $\rho$ closer to 1 for reasoning compute and −1 for accuracy indicate better monotonicity)
> | Models             | Compute↑      |                              |                 |               |      | Log Accuracy↓  |                               |                 |                |      |
> |-----------------|------|---------|-------|------|-----|------|---------|-------|------|-----|
> | |Math|Science| Lang.    | Code     | All  | Math       | Science   | Lang.      | Code       | All  |
> | DeepSeek-R1-7B (reference)   |0.956 |0.975 |0.901 |0.970 |0.991 |-0.946 |-0.876 |-0.899 |-0.818 |-0.978 |
> | Nemontron-14B   |0.964 |0.976 |0.917 |0.970 |0.993 |-0.778 |-0.751 |-0.793 |-0.911 |-0.913 |
> | DeepSeek-R1-14B |0.978 |0.973 |0.903 |0.980 |0.990 |-0.888 |-0.888 |-0.803 |-0.933 |-0.981 |
> | Qwen3-80B-Next  |0.977 |0.984 |0.794 |0.993 |0.992 |**-0.403** |**-0.551** |-0.774 |-0.907 |-0.973 |
> | Sky-T1-32B      |0.988 |0.982 |0.711 |0.874 |0.967 |-0.939 |-0.876 |-0.783 |-0.860 |-0.963 |
>
> **Compositionality Results on LoRe-Compo** ($\mathrm{nMAD}$ closer to 0 indicate better compositionality)
> | Models | nMAD (Compute) ↓ | nMAD (Log Accuracy) ↓|
> |-----------------|------|---------|
> | DeepSeek-R1-7B (reference) | **0.337** | 1.170 |
> | Nemontron-14B | 0.393 | 1.073 |
> | DeepSeek-R1-14B | 0.368 | 1.131|
> | Qwen3-80B-Next | **0.411** | 1.487 |
> | Sky-T1-32B | 0.392 | 1.900 |
>
> **(B) Generalization of SFT-Compo Across Different Model Families**
>
> To further demonstrate the generalization capability of SFT-Compo, we conduct experiments on different model families, including Phi4-mini-reasoning from Microsoft. Our method still achieves consistent gains over all benchmarks and model families.
>
> | Base Model | Method | AIME24 | AIME25 | AMC23 | MATH500 | GSM8K | Olympiad | Pass@1 |
> |-------------------|------------|--------|--------|-------|--------|-------|----------|--------|
> | Phi-4-mini | Base      | 32.5   | 24.6   | 71.9  | 86.4   | 91.4  | 50       | **59.5**   |
> |  | SFT       | 40.0   | 26.2   | 73.4  | 87.0   | 91.6  | 49.5     | **61.3**   |
> |  | SFT-Compo | 43.7   | 31.7   | 76.9  | 87.0   | 92.8  | 51.2     | **63.9**   |

---

> > ### Author Response · Authors · 2025-11-27
> > **Part 2**
> >
> > > **W2**: In SFT-Compo, the training triplets appear to be generated solely using DeepSeek-R1-Distill-Qwen-14B. Have the authors considered incorporating multiple teacher models or alternative data sources to enhance generalization?
> >
> > We thank the reviewer for the insightful suggestions. To further demonstrate the robustness of our conclusions, we **conducted additional experiments** with diverse teacher models and data sources. Specifically, we: (1) used **DeepSeek-R1-Distill-Qwen-7B** as the teacher model to fine-tune the 1.5B model, and (2) employed the **Open-Reasoner-Zero** dataset with DeepSeek-R1-Distill-Qwen-14B as the teacher model to fine-tune both 1.5B and 8B models. All evaluation settings follow those used in our paper to ensure comparability. The results demonstrate that SFT-Compo consistently achieves gains across all benchmarks, regardless of the choice of teacher model or data source.
> >
> >
> > **Additional Evaluation with DeepSeek-R1-Distill-Qwen-7B as the Teacher Model:**
> > | Base Model                 | Method    | AIME24 | AIME25 | AMC23 | MATH500 | GSM8K | Olympiad | Pass@1 |
> > |---------------------------|-----------|--------|--------|-------|---------|-------|----------|--------|
> > | DeepSeek-R1-1.5B | Base      | 18.8   | 20.4   | 59.7  | 71.6    | 81.2  | 33.8     | **47.6**   |
> > |  | SFT       | 21.4   | 22.7   | 61.1  | 74.8    | 82.6  | 36.5     | **49.9**   |
> > |  | SFT-Compo | 22.5   | 23.9   | 65.0  | 78.3    | 82.7  | 38.2     | **51.8**   |
> >
> >
> > **Additional Evaluation with Open-Reasoner-Zero Dataset:**
> > | Base Model                  | Method    | AIME24 | AIME25 | AMC23 | MATH500 | GSM8K | Olympiad | Pass@1 |
> > |-----------------------------|-----------|--------|--------|-------|---------|-------|----------|--------|
> > | DeepSeek-R1-1.5B | Base      | 18.8   | 20.4   | 59.7  | 71.6    | 81.2  | 33.8     | **47.6**   |
> > |  | SFT       | 21.2   | 20     | 58.8  | 77      | 82.6  | 35.5     | **49.2**   |
> > |  | SFT-Compo | 21.3   | 23.3   | 68.8  | 78.6    | 81.7  | 39.1     | **52.1**   |
> > | DeepSeek-R1-8B   | Base      | 28.3   | 22.9   | 71.9  | 76.4    | 86.5  | 40.9     | **54.5**   |
> > |    | SFT       | 27.8   | 23.3   | 73.4  | 79.4    | 84.8  | 44.3     | **55.5**  |
> > |    | SFT-Compo | 30.8   | 25     | 76.9  | 82.7    | 89.4  | 45       | **58.3**   |
> >
> > ***
> >
> > > **W3**: The paper only compares the base models, SFT, and SFT-Compo. While this shows that SFT-Compo improves over the original models, it lacks comparisons with other baseline methods, such as Thinkless or AdaptThink.
> >
> > We address this concern from the following perspectives,
> >
> > - **SFT-Compo is conceptually orthogonal to these baselines.** Thinkless and AdaptThink are models equipped with reasoning length control. In contrast, SFT-Compo is a simple yet effective fine-tuning approach that enforces compositionality, and can be **directly applied on top of any model with reasoning length control**. It is an alternative to standard SFT that can coexist with and potentially further improve these models.
> > - **To further validate the effectiveness of our approach, we conducted additional evaluations on two more baselines**, including Thinkless and AdaptThink. It significantly outperforms all baselines on the 1.5B model by a substantial margin and remains competitive on the 7B model, while using **far fewer** training examples. This demonstrates the effectiveness of enforcing compositionality via SFT-Compo.
> >
> > | Method | Dataset Size | AIME24 | AIME25 | AMC23 | MATH500 | GSM8k | Olympiad | Pass@1 |
> > | :--- | :---: | :---: | :---: | :---: | :---: | :---: | :---: | :---: |
> > | ***DeepSeek-R1-1.5B*** | | | | | | | | |
> > | Base | NA | 18.8 | 20.4 | 59.7 | 71.6 | 81.2 | 33.8 | 47.6 |
> > | SFT | 3.9K | 20.4 | 21.5 | 59.6 | 76.4 | 81.7 | 36.1 | 49.3 |
> > | Thinkless-1.5B | **2.3M** | 26.3 | 20 | 56.6 | 76.8 | 82.3 | 40.6 | **50.4** |
> > | AdaptThink-1.5B | **40K** | 23.3 | 27.1 | 55.9 | 63.6 | 82 | 35.7 | **47.9** |
> > | SFT-Compo | **3.9K** | 26.2 | 21.7 | 65.0 | 77.6 | 85.1 | 38.7 | **52.4** |
> > | ***DeepSeek-R1-7B*** | | | | | | | | |
> > | Base | NA | 36.3 | 27.5 | 79.0 | 86.8 | 91.0 | 48.1 | 61.5 |
> > | SFT | 3.9K | 40.0 | 32.5 | 80.4 | 88.0 | 91.6 | 48.4 | 63.5 |
> > | AdaptThink-7B | **40K** | 41.2 | 30.4 | 77.5 | 90.4 | 92.6 | 52.5 | **64.1** |
> > | SFT-Compo | **3.9K** | 43.3 | 33.2 | 80.6 | 88.8 | 91.6 | 50.5 | **64.7** |

---

### Official Review · Reviewer_YCRJ · 2025-10-30

**Soundness:** 2
**Presentation:** 3
**Contribution:** 2
**Rating:** 2
**Confidence:** 3

**Summary:**

This paper introduces the Laws of Reasoning (LORE), a theoretical framework designed to formalize the reasoning behavior of Large Reasoning Models. The proposed framework contains a compute law and an accuracy law. Specifically, compute should scale proportionally with question complexity and accuracy should decay exponentially with increased complexity.
To evaluate this, the authors create LORE-BENCH. Their evaluation finds that while most LRMs satisfy monotonicity, they fail significantly at compositionality
Finally, the paper proposes SFT-Compo, a fine-tuning method that enforces the compute compositionality law. Experiments show SFT-Compo successfully improves compositionality and boosts performance on general reasoning benchmarks as well.

**Strengths:**

a. This paper is well-written and easy to follow.

b. This paper proposes a theoretical framework focusing on the relationship between problem complexity with accuracy and computation. The authors find that current models exhibit reasonable monotonicity but lack compositionality.

**Weaknesses:**

a. The evaluation is restricted to models with less than 10 billion parameters. It is a critical question whether significantly larger models inherently follow the proposed compositionality law due to emergent capabilities, or if SFT-Compo remains necessary for them.

b. LORE-COMPO (two totally independent questions followed by an arbitrary combination) is rare in natural language. It contradicts with the example in figure 1. The compositional question is that square the answer of the first question. The two question are clearly related while the question in LORE-COMPO are two isolated problems.

c. LORE-COMPO tests compositionality as two totally independent questions. The model has not seen this type of question during training. And this type of compositionality with two independent question is rare in real world scenarios as they can not be written into a complex question similar to what is shown in Figure 1. This makes the scope of this paper somewhat limited.

**Questions:**

What are the hyperparameters used for evaluation with AIME. Such as temperature and the how many runs have you runned on the benchmark?

---

> ### Author Response · Authors · 2025-11-27
> **Part 1**
>
> *Thank you for your constructive and supportive reviews that greatly improve our manuscript! Following your review, we conducted **extensive additional experiments**, including evaluations on four large-scale models, and additional experiments on LoRe-Compo and AIME. We respond below to your questions and concerns, which we have incorporated into the revised version.*
>
> ***
>
> > **W1**: The evaluation is restricted to models with less than 10 billion parameters.
>
> We thank the reviewer for the insightful suggestions. In response, we conducted additional evaluations on **four more large-scale models**, including OpenReasoning-Nemotron-14B, DeepSeek-R1-Distill-Qwen-14B, Sky-T1-32B-Preview, and Qwen3-Next-80B across LoRe-Mono and LoRe-Compo. Our experiments reveal two important findings,
>
> - **All these large-scale models failed to exhibit compositionality.** The $\mathrm{nMAD}$ metric used to evaluate compositionality is large for both reasoning compute and log accuracy across all four models. As shown in Appendix E, most points deviate substantially from the ideal $y=x$ line. This demonstrates that even significantly larger models do not naturally follow the compositionality.
> - **Larger models do not necessarily better satisfy monotonicity or compositionality than small-scale models.** These large-scale models often perform worse on compositionality than the smaller models reported in our paper. For instance, Qwen3-Next-80B, despite being significantly larger, performs worst among these models on compositionality for reasoning compute, with an $\mathrm{nMAD}$ of 0.411 compared to 0.337 for DeepSeek-R1-Distill-Qwen-7B (**a 22% degradation in compositionality**). Qwen3-Next-80B also exhibits the poorest monotonicity across all evaluated models on Math and Science domains, with reasoning accuracy failing to decrease systematically with complexity.
>
> Detailed experiments and visualizations are provided in **Section 3.3 and Appendix E** of the revised paper, which are marked in blue.
>
> **Monotonicity Results on LoRe-Mono** (Spearman correlations $\rho$: $\rho$ closer to 1 for reasoning compute and −1 for accuracy indicate better monotonicity)
> | Models             | Compute↑      |                              |                 |               |      | Log Accuracy↓  |                               |                 |                |      |
> |-----------------|------|---------|-------|------|-----|------|---------|-------|------|-----|
> | |Math|Science| Lang.    | Code     | All  | Math       | Science   | Lang.      | Code       | All  |
> | DeepSeek-R1-7B (reference)   |0.956 |0.975 |0.901 |0.970 |0.991 |-0.946 |-0.876 |-0.899 |-0.818 |-0.978 |
> | Nemontron-14B   |0.964 |0.976 |0.917 |0.970 |0.993 |-0.778 |-0.751 |-0.793 |-0.911 |-0.913 |
> | DeepSeek-R1-14B |0.978 |0.973 |0.903 |0.980 |0.990 |-0.888 |-0.888 |-0.803 |-0.933 |-0.981 |
> | Qwen3-80B-Next  |0.977 |0.984 |0.794 |0.993 |0.992 |**-0.403** |**-0.551** |-0.774 |-0.907 |-0.973 |
> | Sky-T1-32B      |0.988 |0.982 |0.711 |0.874 |0.967 |-0.939 |-0.876 |-0.783 |-0.860 |-0.963 |
>
> **Compositionality Results on LoRe-Compo** ($\mathrm{nMAD}$ closer to 0 indicate better compositionality)
> | Models | nMAD (Compute) ↓ | nMAD (Log Accuracy) ↓|
> |-----------------|------|---------|
> | DeepSeek-R1-7B (reference) | **0.337** | 1.170 |
> | Nemontron-14B | 0.393 | 1.073 |
> | DeepSeek-R1-14B | 0.368 | 1.131|
> | Qwen3-80B-Next | **0.411** | 1.487 |
> | Sky-T1-32B | 0.392 | 1.900 |

---

> ### Author Response · Authors · 2025-11-27
> **Part 2**
>
> > **W2 & W3**: LoRe-Compo is rare in natural language. It contradicts with the example in Fig. 1. The two question in Fig. 1 are clearly related while the question in LoRe-Compo are two isolated problems. This makes the scope of this paper somewhat limited.
>
> We address this concern by clarifying four key points: **(1)** the motivation and effectiveness of enforcing compositionality, **(2)** the inherent challenge of evaluating compositionality directly from natural language, **(3)** the construction principles underlying LoRe-Compo, and **(4)** the relationship between LoRe-Compo and Fig. 1.
>
> - The key motivation of our paper is to **understand the reasoning behaviors of LRMs *in principle* and leverage these insights to guide better model design *in practice*.** Our benchmark serves as a **“debugging tool” for diagnosing** the limitations of existing models. Importantly, **our fine-tuned models**, where such compositionality is enforced, **yield broad and substantial improvements in general reasoning** across six benchmarks. Therefore, the scope of this paper is not limited to a narrow or artificial setting. Notably, SFT-Compo achieves average accuracy improvements of +4.8 on DeepSeek-R1-Distill-Qwen-1.5B and +5.0 on DeepSeek-R1-Distill-Llama-8B with only 3.9K training examples.
> - **Evaluating compositionality involves a fundamental trade-off between how natural the questions are and how suitable they are for rigorously evaluating reasoning laws.** To examine compositionality in a controlled way, we need sub-questions to be independent. However, natural composite questions rarely decompose into independent sub-questions without semantic overlap. Such overlap makes it difficult to attribute deviations to compositionality rather than shared reasoning efforts. Our current design prioritizes rigorous evaluation, representing a principled and scalable first step. The development of more natural but still independent settings remains an important direction for future work.
> - **Though this idealized form of compositionality is rare in practice, the underlying reasoning operation is common in real-world applications**: decomposing problems into subproblems, solving them independently, and integrating their solutions. This includes multi-step mathematical and scientific problem-solving, multi-stage data processing and software pipelines, and embodied tasks requiring sequential navigation and manipulation.
> - We agree with the reviewer that Fig. 1 and LoRe-Compo are different, because **they play complementary roles: Fig. 1 serves as a natural, qualitative observation of abnormal reasoning, whereas LoRe-Compo is a controlled, quantitative experiment designed to rigorously evaluate our reasoning laws.** Together, they provide both motivating evidence (Fig. 1) and rigorous evaluation (LoRe-Compo) for the necessity of SFT-Compo. We thank the reviewer for this valuable feedback and will modify Fig. 1 to avoid confusion.
>
> ***
>
> > **W2 & W3**: LoRe-Compo tests compositionality as two totally independent questions. The model has not seen this type of question during training.
>
> We thank the reviewer for raising a valid concern about the potential distribution shift induced by the new question format. To investigate this, we **analyzed reasoning accuracy for each model on LoRe-Compo and found that distribution shift is not the primary challenge**.
>
> Although the model has not seen this “two independent questions concatenated” format during training, models maintain reasonably high performance on the composite questions. For example, DeepSeek-R1-Distill-Llama-8B achieves **83.3%** accuracy on LoRe-Compo composite questions while failing to exhibit compositionality. **It indicates that the construction itself does not fundamentally hinder their problem-solving capabilities.** What LoRe-Compo reveals instead is an abnormal pattern in how the reasoning compute is allocated.
>
> **Accuracy on Composite Questions in LoRe-Compo:**
> | Models                          | Composite Accuracy |
> |--------------------------------|-----------:|
> | DeepSeek-R1-1.5B               | 35.4 |
> | Thinkless-1.5B                 | 78.1 |
> | DeepSeek-R1-7B                 | 89.4 |
> | AdaptThink-7B                  | 90.2 |
> | Phi-4-mini                     | 87.5 |
> | DeepSeek-R1-8B                 | **83.3** |
> | Qwen3-80B-Next                 | 96.1 |
> | DeepSeek-R1-14B                | 91.4 |
> | OpenReasoning-Nemotron-14B     | 87.7 |
> | Sky-T1-32B                     | 74.4  |

---

> > ### Author Response · Authors · 2025-11-27
> > **Part 3**
> >
> > > **Q1**: What are the hyperparameters used for evaluation with AIME. Such as temperature and the how many runs have you runned on the benchmark?
> >
> > In our original paper, we sampled 8 runs per model with a temperature of 0.6, following the settings in official technical reports. To demonstrate the robustness of our conclusions, we **conducted additional experiments with 32 runs** on two representative models, and our findings remain fully consistent with those reported in the paper.
> >
> > **Pass@1 with 32 runs:**
> > | Base Model | Method | AIME24 | AIME25 |
> > | :--- | :--- | :---: | :---: |
> > | **Deepseek-R1-1.5B** | Base | 17.7 | 20.3 |
> > | | SFT | 18 | 21.2 |
> > | | SFT-Compo | 23.1 | 22.2 |
> > | **Deepseek-R1-8B** | Base | 29.7 | 22.6 |
> > | | SFT | 30.0 | 25.0 |
> > | | SFT-Compo | 33.2 | 27.9 |
> >
> > **Table 3 with 8 runs (for reference):**
> > | Base Model | Method | AIME24 | AIME25 |
> > | :--- | :--- | :---: | :---: |
> > | **Deepseek-R1-1.5B** | Base | 18.8 | 20.4 |
> > | | SFT | 20.4 | 21.5 |
> > | | SFT-Compo | 26.2 | 21.7 |
> > | **Deepseek-R1-8B** | Base | 28.3 | 22.9 |
> > | | SFT | 30.4 | 24.2 |
> > | | SFT-Compo | 31.3 | 29.2 |

---

### Official Review · Reviewer_eBzN · 2025-10-31

**Soundness:** 3
**Presentation:** 3
**Contribution:** 3
**Rating:** 6
**Confidence:** 3

**Summary:**

This paper aims to understand the reasoning behaviour of large reasoning models. First, the authors focus on the incompatibility between the compute used and the complexity of the question (defined as Turing machine operation steps). I.e. sometimes models use more compute than needed for simple questions, yet generating suboptimal results. This should be related to the 'over thinking' problem that has been studied by the field recently.
In addition, the authors also study an accuracy law.

Since question complexity is hard to quantify, the authors seek to use two surrogates, monotonicity and compositionality, to examine the hypotheses (compute should be linearly related to question complexity). A benchmark is therefore proposed to quantify thee two tractable properties.

Finally, the authors show that most models are reasonable in terms of monotonicity, while lacking compositionality. To this end, the authors proposed a finetuning method to enforce compute-law compositionality.

**Strengths:**

This paper investigates a practically important problem: How the compute and accuracy of LLM vary with question complexity. In addition, the authors not only evaluate the existing methods, but also propose a novel finetuning method to advance the existing models.

The presentation is clear, and the figures and tables are well organized and visually appealing.

**Weaknesses:**

The complexity construction of the data is not very consistent with the definition of complexity in the method part. I would say that the Turing machine steps seem to be a more reasonanle way to define the complexity. While the number of steps used in the datasets requires more justification.

**Questions:**

1. The question complexity is defined based on the concept of Turing machine. During dataset curation, how to justify that the complexity concept used when selecting questions aligns with the complexity concept based on Turing machine steps? For example, the number of matrix operation is used in dataset construction, while this may not be linearly correlated with the Turing machine steps. Otherwise, if

2. In dataset construction, does each matix operation uses same compute? Different matrix operation may require different compute.

3. In addition to matrix operation, does other 'steps' use same compute. If not, then a question requiring more steps may actually need less compute than a question requiring more steps.

---

> ### Author Response · Authors · 2025-11-27
> **Part 1**
>
> *Thank you for your constructive and supportive reviews that greatly improve our manuscript! We respond below to your questions and concerns on **the correlation between the definition of complexity and the constructed complexity in LoRe-Mono**, which we have incorporated into the revised version.*
>
> ***
>
> > **W1 & Q1**: The complexity construction of the data is not very consistent with the definition of complexity. During dataset curation, how to justify that the complexity concept used when selecting questions aligns with the complexity concept based on Turing machine steps? For example, the number of matrix operation is used in dataset construction, while this may not be linearly correlated with the Turing machine steps.
>
> All the concerns the reviewer raised relate to the correlation between the definition of complexity and the constructed question complexity (variant index in Fig. 3) in LoRe-Mono. We clarify this from the following perspectives,
>
> - **LoRe-Mono is designed to evaluate the *monotonicity* of current LRMs, *not their linearity*.** Therefore, the number of matrix operations does not need to correlate linearly with Turing machine steps.  For monotonicity,  it is sufficient to ensure that the complexity of variant $n+1$ is greater than the complexity of variant $n$ ($n≥1$) and does not require each step to use the same compute. For example, in Fig. 3, variant $n+1$ adds one more matrix operation on top of variant $n$, which guarantees an increase in complexity.
> - **A key benefit of introducing monotonicity and compositionality instead of studying reasoning laws directly is that they can be verified without requiring the exact value of complexity $\kappa(x)$.** We agree with the reviewer that complexity can be well defined via a fixed deterministic Turing machine. However, quantifying the exact complexity value $\kappa$ is inherently difficult in practice. To address this challenge, we adopt monotonicity and compositionality as empirical proxies for studying the reasoning laws. While monotonicity relies only on relative comparisons between questions, compositionality tests whether compute is additive over independent question pairs. They are **easy to quantify compared to the definition given by a deterministic Turing machine and are theoretically sufficient to imply our proposed laws** (Proposition 1 & 2). Based on these two properties, we construct LoRe-Mono and LoRe-Compo, respectively.
>
> ***
>
> > **Q2 & Q3**: In dataset construction, does each matix operation uses same compute? Different matrix operation may require different compute. In addition to matrix operation, does other 'steps' use same compute?
>
> These questions follow the same principle as those discussed above, and to summarize, one of our key contributions is to specially design the monotonicity and compositionality properties to study the reasoning laws without requiring exact complexity characterization. Each matrix operation is **not assumed or required to use the same amount of compute**, and the same applies to other questions in the LoRe-Mono constructions.
>
> We would thank the reviewer for raising this insightful discussion, and we have **modified Section 3.1** to prevent potential confusion, with revisions marked in blue.

---

### Author Response · Authors · 2025-12-03
**The Summary Message to New AC (Part 1)**

Dear New Area Chair,

Thank you very much for your time and effort in handling a new batch of submissions. We truly appreciate your work during this busy period. We are also sincerely grateful for the reviewers’ constructive feedback. Before the OpenReview bug, the reviewers on our paper have not had the opportunity to respond to our rebuttal. To support your evaluation, we respectfully summarize reviewers' questions and the key updates and clarifications we made below.

***

We are encouraged to see that several reviewers recognized the **essential contributions** of our paper, including:

- **A principled foundation for analyzing LRM behaviors.** LoRe offers a principled framework for understanding LRMs via the compute and accuracy laws, with monotonicity and compositionality as practical proxies that effectively bridge theoretical hypotheses and empirical evaluation. (Reviewer `eBzN`, `pUQ6`, `HnMv`, `YCRJ`)
- **A simple yet effective fine-tuning approach to enforce reasoning laws.** SFT-Compo is a simple yet effective method that enforces compositionality and significantly enhances general reasoning performance. (Reviewer `eBzN`, `HnMv`)
- **The paper is clearly written and well-organized.** (Reviewer `eBzN`, `YCRJ`)

***
During the rebuttal period, we have conducted **extensive additional experiments** to support our conclusions,

- **Evaluations on four large-scale models.** To address reviewers' concerns about `the generalization of our claims to larger models`, we evaluate four more large-scale models and reveal two important findings: (1) All four large-scale models failed to exhibit compositionality, and (2) Larger models do not necessarily better satisfy monotonicity or compositionality than small-scale models. These two findings **further support and strengthen our claim that current LRMs fail to exhibit compositionality.** Detailed experiments and visualizations are provided in Section 3.3 and Appendix E of the revised paper, which are marked in blue. (Reviewer `YCRJ`, `HnMv`, `pUQ6`)
- **Generalization of SFT-Compo on different model families.** We conduct experiments on Phi4-mini-reasoning from Microsoft. Our method still **achieves consistent gains over all benchmarks** and model families. (Reviewer `HnMv`, `pUQ6`)
- **More robust evaluations on AIME24 and AIME25.** In response to reviewers' questions about our `evaluation hyperparameters`, we conducted supplementary experiments with a larger sampling number of 32 runs for all benchmarks. The results remain **fully consistent** with those reported with 8 runs in the paper. (Reviewer `YCRJ`)
- **Fine-tuning with multiple teacher models and alternative data sources.** While the reviewer is concerned that `fine-tuning with a single teacher model may limit data diversity`, we conduct experiments with diverse teacher models (including DeepSeek-R1-Distill-Qwen-7B) and data sources (including the Open-Reasoner-Zero dataset). The results demonstrate that SFT-Compo **consistently achieves gains** across all benchmarks, **regardless of the choice of teacher model or data source.**  (Reviewer `HnMv`)
- **Baseline comparisons for Thinkless and AdaptThink.** To address the reviewer's concern about `lacking baseline comparisons`, we conducted additional evaluations comparing SFT-Compo with Thinkless and AdaptThink. The results show that SFT-Compo significantly **outperforms all baselines** using **far fewer** training examples.  We further clarify that SFT-Compo is conceptually orthogonal to the baselines the reviewer mentioned and can be directly applied on top of any model with reasoning length control. (Reviewer `HnMv`)
- **Robustness of compositional enforcement to domain shifts.** To address the reviewer's concern about `robustness under domain shift`, we evaluate SFT-Compo models trained on DeepScaler and evaluate them directly on CommonsenseQA (a commonsense reasoning benchmark). These results support that SFT-Compo **improved compositionality even under domain shift.** (Reviewer `pUQ6`)

---

> ### Author Response · Authors · 2025-12-03
> **The Summary Message to New AC (Part 2)**
>
> We also **clarified several key points** to address the reviewers' concerns,
>
> **Most of these comments from Reviewer `eBzN`, `HnMv`, and `pUQ6s` highlight aspects that benefit from clearer explanation rather than fundamental revisions to the proposed approach.**
>
> - Reviewer `eBzN` raises a valid question about the correlation between the definition of complexity and the constructed complexity in LoRe-Mono, and we have **modified Section 3.1 to prevent potential confusion**.
> - The concerns raised by Reviewer `HnMv` are fully addressed and **resolved through our additional experiments**.
> - For Reviewer `pUQ6s` concerns about generalization across broader reasoning domains and multimodal contexts (W1 & Q2), we **provide justification through LoRe's formulation and concrete examples** from visual and embodied reasoning to illustrate why the compute and accuracy laws extend to broader domains. We have also **conducted the analyses requested** by Reviewer `pUQ6s`, including qualitative case studies (W3), a discussion on why enforcing compositionality enhances reasoning capabilities (W4), and clarification on our operational definition of question independence via disjoint concept sets (W5).
>
> **Issues raised by Reviewer `YCRJ`.**
>
> - Reviewer `YCRJ` brings up a question about one component of our benchmark, LoRe-Compo and Fig. 1 in our paper. We address this concern by **clarifying four key points**: **(1)** the motivation and effectiveness of enforcing compositionality, **(2)** the inherent challenge of evaluating compositionality directly from natural language, **(3)** the construction principles underlying LoRe-Compo, and **(4)** the relationship between LoRe-Compo and Fig. 1.
> - We additionally examine whether the new question format introduces a distribution shift. From our experiments, we analyzed reasoning accuracy for each model on LoRe-Compo and found that **the construction itself does not fundamentally hinder their problem-solving capabilities**, indicating that distribution shift is not the primary issue.
>
> ***
> We hope this summary is helpful for your assessment of our submission. We deeply appreciate your time, consideration, and support throughout this process.
>
> Warm regards, \
> Authors of Submission

---

### Meta-Review · Area_Chair_DDvm · 2026-01-06

**Summary:**

This paper proposes the Laws of Reasoning (LoRe) as a conceptual framework to characterize intrinsic reasoning patterns in Large Reasoning Models (LRMs). The authors propose a "compute law," hypothesizing that reasoning computation scales linearly with question complexity, operationalized via monotonicity and compositionality.

Reviewers generally found the topic timely and the presentation clear, and acknowledged the attempt to formalize reasoning behavior and the empirical gains from SFT-Compo. However, concerns centered on the conceptual validity and practical relevance of the proposed proxies (especially compositionality), the initial limited scope of the evaluation, and whether the benchmark accurately reflects the intended notion of complexity and real-world reasoning. While the rebuttal strengthened the empirical side, lingering doubts remain regarding the framework's naturalness, scope, and theoretical grounding.

**Reviewer Concerns:**

Concerns addressed by the rebuttal:
- Limited model scale and generalization (YCRJ, HnMv, pUQ6): Additional experiments on larger (up to 80B) and different model families are shown.
- Single-teacher and baseline comparisons (HnMv): Experiments with multiple teacher models, alternative datasets, and comparisons against Thinkless/AdaptThink are shown.
- Evaluation details and robustness (YCRJ): Clarification of AIME hyperparameters and additional runs are provided.

Remaining concerns:
- Conceptual naturalness and scope of compositionality (YCRJ): Despite justification, LoRe-Compo’s construction using concatenated independent questions remains artificial and weakly aligned with common natural-language compositional reasoning.
- Theoretical justification (pUQ6): The paper still lacks a formal or deeper theoretical argument explaining why enforcing compositionality should universally improve reasoning, beyond intuition and empirical evidence.
- Benchmark representativeness (pUQ6): Reliance on a small number of seed questions and heuristic notions of independence limits confidence in the generality of the conclusions.

**Reviewer Scores:**

While additional experimental results may have slightly increased the original scores for some reviewers, the major remaining concerns described above remain critical and unresolved. Overall, the inherent limitations of this work may outweigh its strengths in its current form.

---

### Decision · Program_Chairs · 2026-01-26

Reject